# Multiplexed Cas9 targeting reveals genomic location effects and gRNA-based staggered breaks influencing mutation efficiency

Santiago Gisler [1], Joana P. Gonçalves [2,3], Waseem Akhtar[1], Johann de Jong[3,4], Alexey V. Pindyurin [5,6], Lodewyk F.A. Wessels[2,3] & Maarten van Lohuizen[1]

Understanding the impact of guide RNA (gRNA) and genomic locus on CRISPR-Cas9 activity is crucial to design effective gene editing assays. However, it is challenging to profile Cas9 activity in the endogenous cellular environment. Here we leverage our TRIP technology to integrate ~ 1k barcoded reporter genes in the genomes of mouse embryonic stem cells. We target the integrated reporters (IRs) using RNA-guided Cas9 and characterize induced mutations by sequencing. We report that gRNA-sequence and IR locus explain most variation in mutation efficiency. Predominant insertions of a gRNA-specific nucleotide are consistent with template-dependent repair of staggered DNA ends with 1-bp 5′ overhangs. We confirm that such staggered ends are induced by Cas9 in mouse pre-B cells. To explain observed insertions, we propose a model generating primarily blunt and occasionally staggered DNA ends. Mutation patterns indicate that gRNA-sequence controls the fraction of staggered ends, which could be used to optimize Cas9-based insertion efficiency.

[1] Division of Molecular Genetics, Oncode and The Netherlands Cancer Institute, Plesmanlaan 121, Amsterdam 1066 CX, The Netherlands. [2] Department of Intelligent Systems, Delft University of Technology, Van Mourik Broekmanweg 6, Delft 2628 XE, The Netherlands. [3] Division of Molecular Carcinogenesis, Oncode and The Netherlands Cancer Institute, Plesmanlaan 121, Amsterdam 1066 CX, The Netherlands. [4] Data & Translational Sciences Group, UCB Biosciences GmbH, Alfred-Nobel-Straße 10, Monheim am Rhein 40789, Germany. [5] Institute of Molecular and Cellular Biology, Siberian Branch of Russian Academy of Sciences, Acad. Lavrentiev Ave. 8, Novosibirsk 630090, Russia. [6] Division of Gene Regulation, Oncode and The Netherlands Cancer Institute, Plesmanlaan 121, Amsterdam 1066 CX, The Netherlands. These authors contributed equally: Santiago Gisler, Joana P. Gonçalves, Waseem Akhtar. Correspondence and requests for materials should be addressed to L.F.A.W. (email: l.wessels@nki.nl) or to M.v.L. (email: m.v.lohuizen@nki.nl)

Genome engineering has seen considerable progress in recent years, nurtured by the emergence of precision editing tools based on the bacterial clustered regularly interspaced short palindromic repeats (CRISPR)-associated protein 9. The CRISPR-Cas9 system complexes the endonuclease enzyme Cas9 with a guide RNA to induce double-strand breaks (DSBs) at a specific DNA locus[1–5]. For target DNA recognition and binding, CRISPR-Cas9 requires the presence of a short, conserved sequence known as protospacer-adjacent motif (PAM). The PAM consists of nucleotides NGG and is located downstream of the target sequence[6,7].

Cas9-induced DSBs activate the cellular DNA damage response, mainly through non-homologous end-joining (NHEJ) or homology-directed repair (HDR)[8]. NHEJ is the most common DNA repair pathway[9]. In NHEJ, DNA ends are processed independently without a template prior to ligation, often producing mutations at the break site. HDR relies on sequence homology for repair and therefore depends on the availability of a donor DNA template, which can be acquired from the sister chromatid in S-phase[8]. In genome engineering applications, error-prone repair of Cas9-induced DSBs can be exploited to disrupt the target sequence and generate gene knockouts. Exogenous genetic material can also be integrated into host DNA by providing repair templates with custom oligonucleotides flanked by homology arms.

The CRISPR-Cas9 technology is used extensively for gene editing in vitro and in vivo, yet most factors controlling its nuclease activity are poorly understood. While effects of guide RNA on Cas9 nuclease efficiency and target specificity have been extensively characterized[10–14], the influence of target sequence on induced mutation patterns remains unclear. Little is also known on the impact of genomic and epigenomic context at the target locus[15]. Early studies found that chromatin accessibility or DNA methylation affect the binding of catalytically inactive Cas9 (dCas9)[16–18]. Others showed that Cas9 binding and cleavage are sensitive to chromatin changes induced by nucleosome occupancy[19,20] or administration of doxycycline[21]. Most literature suggests that genomic context influences Cas9 binding and cleavage, but effects on editing efficiency are less well understood. Several studies have observed weak correlations between epigenomic context and Cas9-induced mutation frequency at endogenous targets[17,22–24]. In particular, two of these studies showed that the significant effect of epigenomic context on Cas9 binding did not necessarily result in a detectable effect on Cas9-induced mutation frequency[17,22].

Here, we characterize Cas9-induced mutations in the genomes of mouse embryonic stem (mES) cells. We aim to survey many loci for sufficient statistical power with minimal disruption of the native environment. However, it is not trivial to scale up the number of endogenous Cas9 targets. Compromising on guide RNA specificity increases off-target effects. Targeting repetitive sequences creates challenges for alignment and mutation calling, and results might not generalize to other kinds of sequences. Both approaches can generate a large number of cleavage events per cell, eventually leading to genomic instability and unreliable Cas9 activity profiling as a result. While multiple guides could instead be used to expand Cas9 targeting, this would also introduce target heterogeneity. Alternatively, we integrate thousands of barcoded target sequences throughout the genomes of a population of mES cells using our TRIP technology[25,26]. In this way, we multiplex Cas9 cleavage while keeping the number of targets per cell under control. The use of TRIP reporters further enables sequence-independent analysis of effects across the targeted loci. We investigate the usefulness of these hybrid exogenous-endogenous loci to profile Cas9 activity, and assess the impact of guide RNA sequence and targeted locus on induced mutation frequency and patterns.

## Results

**RNA-guided Cas9 targeting of integrated reporters**. We profiled CRISPR-Cas9-induced mutations across the genomes of mES cells. First, we used TRIP[25,26] to embed barcoded reporter genes randomly throughout the host DNA (Fig. 1a). We established a clonal TRIP cell line containing 36 PGK-driven integrated reporters (IRs) per cell, and a multi-promoter TRIP pool with ~1k IRs distributed heterogeneously across cells (Fig. 1b). We designed three single-guide RNAs (sgRNAs) targeting sites near the 3′-end of the IR gene body, cloned them into Cas9-sgRNA plasmids and used them in independent assays (Fig. 1c). After selecting Cas9-sgRNA-carrying cells, we amplified and sequenced IR target regions to characterize induced lesions. As proof-of-concept, we performed Cas9 disruption assays using sgRNA1-3 in the TRIP cell line (Fig. 1c). By disruption, we refer to cleavage without the use of exogenous DNA. We also did disruption assays in TRIP pools to study mutations at a large number of loci. In addition, we performed editing involving the knock-in of a 21-nt single-stranded oligodeoxynucleotide (ssODN) to characterize template-dependent insertions. We used both sgRNA2/3 for disruption in TRIP pools (-ssODN), and only sgRNA2 for editing ( + ssODN) since the proximity between the sgRNA3 target and the IR barcode prevented the design of proper homology arms. Finally, we analyzed all 36 IRs in the cell line, and also the 1359 IRs with at least 30 reads in all pool assays.

**Variation in Cas9-induced mutation frequency across IR loci**. We first analyzed Cas9-induced mutation frequencies at the targeted loci, and the effects of different factors on those frequencies. Mutation frequency was determined per IR as the fraction of reads carrying a mutation amongst all reads mapped to that specific IR. Overall, Cas9-targeted IR sequences showed high mutation frequencies genome-wide in both cell lines and pools. Cell line averages were ~50% for sgRNA1/2 and ~65% for sgRNA3, while pools reached ~30% for sgRNA1/2 and ~60% for sgRNA3 (Figs. 2a, b). We examined how Cas9-induced mutation frequencies varied with sgRNA, IR locus, ssODN, and promoter.

**Guide RNA sequence-driven variation in mutation frequency**. In line with previous reports, different guide RNAs led to systematic variation in IR mutation frequencies[1,13,27]. Guide sgRNA3 was most efficient in the cell line with average 1.30-fold and 1.26-fold increases in mutation frequency relative to sgRNA1/2 (effect sizes 15% ± 1% for sgRNA3 vs. sgRNA1 and 13% ± 1% for sgRNA3 vs. sgRNA2, both with $p \approx 2.91 \times 10^{-11}$, two-tailed Wilcoxon rank sum tests, Fig. 2a). The difference was largely due to insertions, showing 12.5-fold and 5.26-fold increases with sgRNA3 compared to sgRNA1/2. Deletion frequency was more comparable across guides, varying by 1.23-fold and 1.10-fold between sgRNA3 and sgRNA1/2. Guide sgRNA3 was also the most efficient in TRIP pools, promoting an average 1.96-fold increase in mutation frequency (effect size 30% ± 1% sgRNA3-ssODN vs. sgRNA2-ssODN, $p < 2.20 \times 10^{-16}$, two-tailed Wilcoxon signed rank test; Figs. 2b, c). Again, insertions increased by 6.76-fold while deletions varied by 1.45-fold. Overall, we found that sgRNA1-3 resulted in different levels of mutation efficiency, mostly contributed by insertions.

**Locus-associated variation in mutation frequency**. We questioned whether certain IR loci would be more prone to mutations than others. We saw that IR mutation frequency correlated strongly across cell line assays using the three sgRNAs, with $R^2 > 0.85$ and

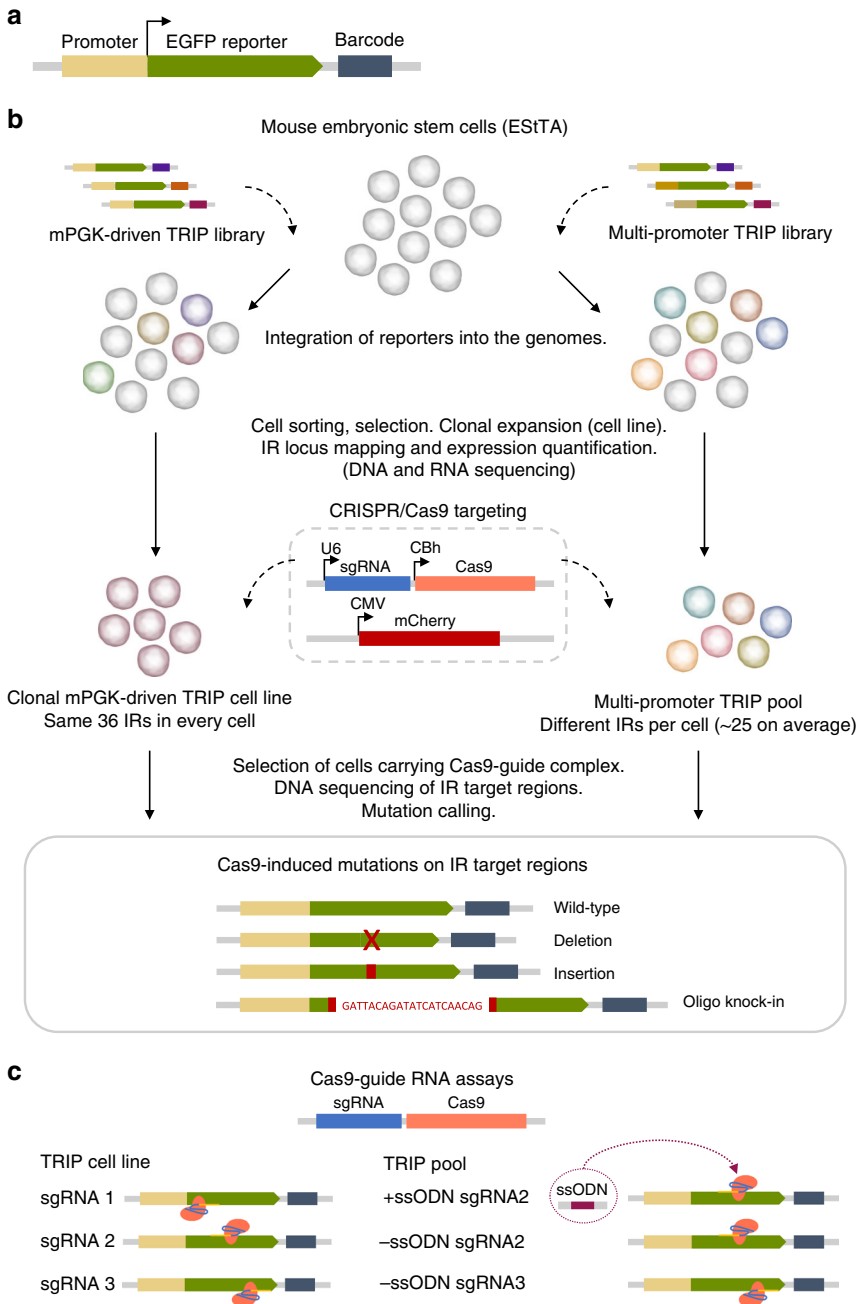

**Fig. 1** Overview of CRISPR-Cas9 assays in TRIP cell line and pools. **a** Barcoded TRIP reporter construct. **b** Clonal PGK-driven TRIP cell line with 36 IRs (left), and TRIP pool containing ~ 1k IRs with various promoters (right) - CMV, cMyc, Hoxb1, Nanog, Oct4, p53, PGK. Genomic location and expression of IRs were determined by DNA and RNA sequencing prior to Cas9 targeting of IR regions using different guides. Targeted DNA sequencing of IR regions was further used to characterize mutations arising from repair of Cas9-induced DSBs. (**c**) Cas9-guide RNA combinations used in independent assays. TRIP cell line was targeted using Cas9 complexes with sgRNA1, sgRNA2 or sgRNA3 (left). In TRIP pool assays, Cas9 was complexed with sgRNA2 or sgRNA3 (right). Knock-in of a single-stranded oligodeoxynucleotide (ssODN) was performed with sgRNA2

F-test $p < 3.13 \times 10^{-16}$ (Fig. 2a), and across TRIP pool assays with $R^2 > 0.72$ and F-test $p < 2.16 \times 10^{-16}$ (Fig. 2d). Correlations were lower for insertions ($R^2 \leq 0.30$) than deletions ($R^2 \geq 0.72$), likely due to the scarcity of insertion events. The highly reproducible mutation frequencies revealed consistent locus-specific susceptibility to Cas9-induced mutations (Figs. 2a, d).

**Knock-in and error-based insertion frequency**. We examined the frequencies of error-based and knock-in insertions. Most insertions induced by Cas9-sgRNA2 in the TRIP pool resulted from errors in endogenous repair of Cas9-induced DSBs, with

average frequencies per IR of 3.78% and 3.24% in disruption and editing experiments respectively (Fig. 2b). Knock-ins occurred only in editing assays with the integration of the designed 21-nt ssODN at the break site by HDR. Knock-in efficiency was low, as expected[15], with an average of 1.74% per IR (Fig. 2b, left plot). However, knock-ins were more frequent than error-based insertions at IRs with total mutation frequency larger than 70% (Fig. 2e).

**Effects on mutation frequency**. To quantify the effect of the above factors on IR mutation frequency, we modelled mutation

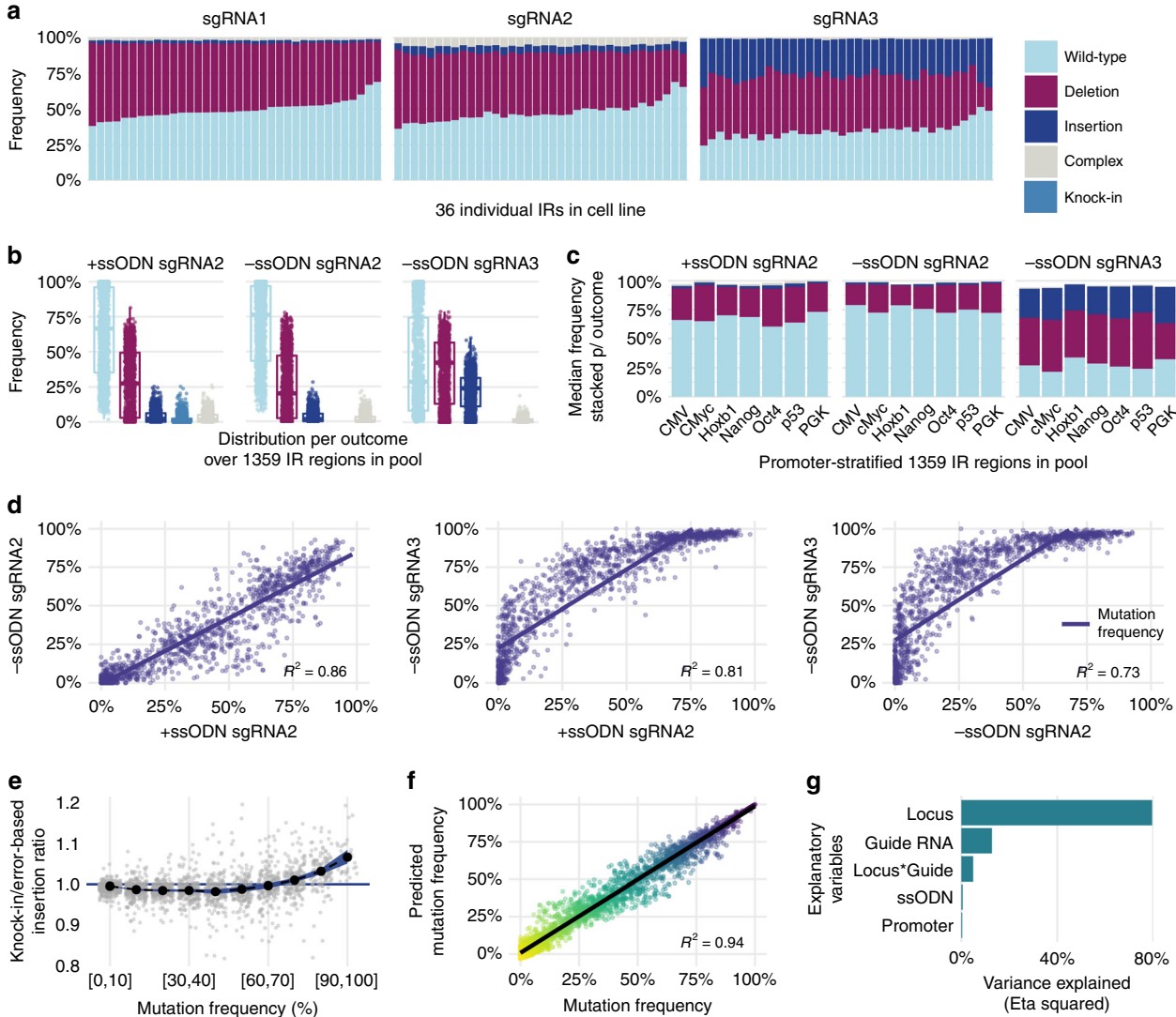

**Fig. 2** Contribution of IR locus, guide RNA, promoter and ssODN to Cas9-induced mutation frequency. **a** Frequency per outcome in cell line Cas9 assays, showing effects of IR locus and guide RNA. Each bar represents one of 36 IRs in the cell line, and each colored band denotes the fraction of reads exhibiting a particular outcome among all reads mapped to such IR (vertical axis). Outcomes: wild-type in light blue, deletion in red, insertion in dark blue, and complex mix of mutations in beige. **b** Frequency per outcome in TRIP pool assays, for guide RNA and ssODN inclusion combinations. Dots denote frequency (vertical axis) per outcome (color) for 1359 IRs with at least 30 reads in all assays. Boxes show the median, first and third quartiles of the frequency distributions; whiskers extend to 1.5 times the inter-quartile range from the top and bottom of the box. **c** Frequency per outcome in TRIP pools, stratified by promoter. Each bar denotes the subset of IRs associated with a given promoter; colored bands denote median frequency per outcome. **d** Correlation of IR mutation frequency across TRIP pool assays. Each dot indicates mutation frequency of a given IR in two different experiments (horizontal and vertical axes). Linear regression lines and corresponding R[2] values denote correlations. **e** Ratio between knock-in and error-based insertions (vertical axis) with respect to binned IR mutation frequency (horizontal axis). Grey dots indicate ratios for individual IRs, black dots denote mean ratios within bins, blue ribbon shows 0.95 confidence interval around the mean. **f** Goodness-of-fit of linear regression model predicting mutation frequency based on IR locus, guide RNA, ssODN, promoter, and (locus, guide) interaction term. **g** Effect size or variance explained by variables in the regression model. Plotted are eta squared values for multi-way ANOVA tests based on type II sum of squares. Source data are provided in the Source Data file

frequencies in the TRIP pool as a linear function of IR locus, guide RNA, ssODN inclusion, promoter, and an interaction term for the joint (non-additive) contribution of locus and guide (Figs. 2f, g). The linear regression model yielded a goodness-of-fit of $R^2 \approx 0.98$ (Fig. 2f). Using multi-way ANOVA tests, we determined the effect size of each factor in the model (Fig. 2g). IR locus explained ~79.5% of the variation in mutation frequency, while guide RNA sequence was responsible for ~12.7%, and locus-guide interaction accounted for ~4.9%. ssODN and promoter had negligible effect, with less than 1% together. These results confirmed that IR locus and guide RNA are major determinants of mutation frequency. Note that IR locus encapsulates a variety of

factors that make a locus unique, including genomic context or the interaction between an IR and the host DNA.

**Association between genomic context and mutation frequency.** We sought to analyze the contribution of genomic context at IR loci to Cas9-induced mutation frequencies. Specifically, we examined the relation between the mutation frequency for IRs in TRIP pools (Fig. 3a) and transcriptional, genomic and epigenomic (TGE) features (Figs. 3b–d). Transcriptional features included IR expression in our TRIP cells, and gene expression and transcription factor binding in wild-type mES cells[28]. Genomic features comprised metrics such as GC content and

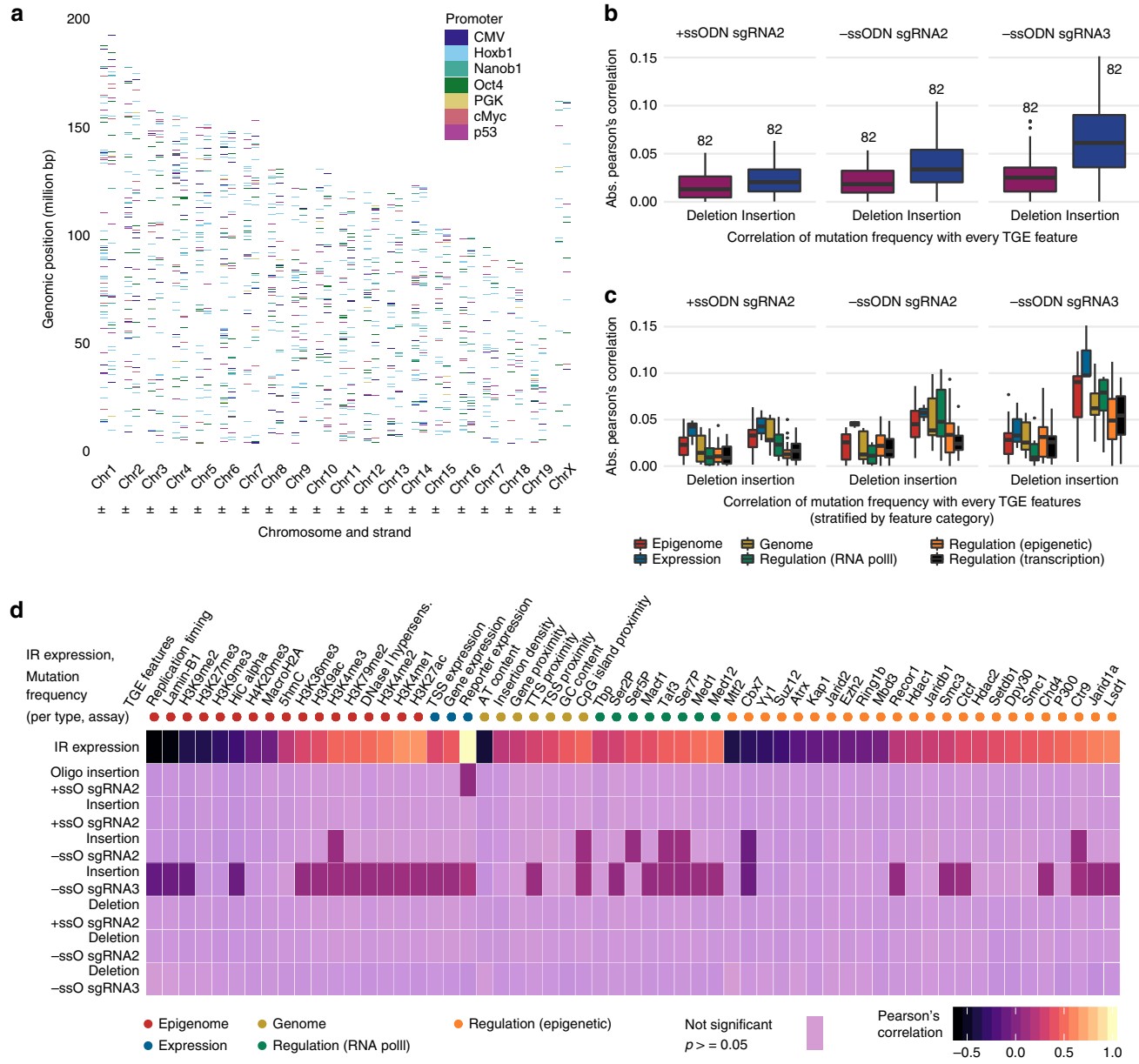

**Fig. 3** Correlation of TGE features with Cas9-induced IR mutation frequency in the TRIP pool. **a** Genomic location of the 1359 IRs with at least 30 mapped reads in all TRIP pool Cas9 assays. Each tick denotes the location of an IR on the chromosome, colored according to the associated promoter. **b** Correlation of TGE features with IR mutation frequency per guide RNA. Boxplots show the distribution of absolute Pearson's correlations between deletion (red) or insertion (blue) frequency and each of 82 distinct TGE features across IRs. Boxes show the median, first and third quartiles of the frequency distributions; whiskers extend to 1.5 times the inter-quartile range from the top and bottom of the box. **c** Correlation of IR mutation frequency with TGE features stratified per category. Boxplots show the distribution of absolute Pearson's correlations between deletion or insertion frequency and each of 82 TGE features stratified into six categories (color-coded according to legend). **d** Correlation between IR expression or IR mutation frequency and TGE features. Heatmap shows the Pearson's correlation between IR expression or IR mutation frequency (deletion or insertion) in the different TRIP pool assays (rows), and individual TGE features from a subset of 62 (columns), including all except transcriptional regulators without known epigenetic activity. Cells are gradient-colored based on correlation values, and color intensity denotes significance of adjusted p-value. Colored circles at the top indicate TGE feature categories. Source data are provided in the Source Data file

gene proximity in mES cells[28]. Epigenomic features included chromatin density from Hi-C assays, and chromatin immuno-precipitation (ChIP) data for numerous histone modifications, DNaseI hypersensitivity, and Lamin-B1 in mES cells[28].

We quantified TGE features within a region of 2 kb surrounding each IR locus and calculated their correlation with IR mutation frequency. Mutation frequency correlated weakly with TGE features (Pearson's $r = [-0.123, 0.151]$, (Fig. 3b–d)), but insertions showed stronger association than deletions (effect size

$0.021 \pm 0.007$, $p \approx 1.91 \times 10^{-9}$, two-tailed Wilcoxon signed rank test sgRNA2 ± ssODN and sgRNA3-ssODN insertions vs. sgRNA2 ± ssODN and sgRNA3-ssODN deletions). Correlations were also higher for sgRNA3 compared with sgRNA2, more so for insertions than deletions (effect size $0.022 \pm 0.008$, $p \approx 2.67 \times 10^{-7}$ for insertions, $0.016 \pm 0.009$, $p \approx 0.0001$ for deletions, two-tailed Wilcoxon signed rank test sgRNA3-ssODN vs. sgRNA2-ssODN). We investigated whether changes induced by TRIP IRs could have disrupted the correlation between IR mutation

frequency and expression-based features. For this we analyzed IR expression, quantified as IR barcode counts in cDNA normalized by IR barcode counts in genomic DNA obtained by high-throughput sequencing of a 168 bp region spanning the Cas9 target sites in TRIP mES cells. The correlation with mutation frequency was similarly weak for IR expression than for expression in wild-type mES cells, and we therefore excluded a potential impact of TRIP on these results (Fig. 3d, Supplementary Figure 3). Also reassuring was the fact that IR expression correlated strongly with wild-type TGE features (Pearson's $r = [-0.603, 0.648]$, Fig. 3d). Additionally, we looked into translocations upon Cas9 targeting as an eventual source of variation in mutation frequency, and found no evidence that they played a role in this regard (Supplementary Tables 1 and 2).

Expression of IRs and endogenous genes showed the highest association with mutation frequency amongst six categories of TGE features (Fig. 3c). Moreover, features related to the RNA PolII complex involved in genome-wide transcription initiation produced larger effects than transcription factors. Individually, phosphorylated RNA PolII (Ser2P, Ser5P, Ser7P) and influencers of transcription initiation such as Ctr9, Taf3 and CpG islands correlated positively with insertion frequency (Fig. 3d). We saw similar effects for H3K4 methylation and other histone modifications (H3K9ac, H3K27ac, H3K36me3, H3K79me2) associated with actively transcribed regions[29]. Correlating negatively with insertion frequency were known indicators of transcription inhibition, namely: lamina-associated domains (Lamin-B1), heterochromatin mark H3K9me2, and the Polycomb Repressive Complex 1 (PRC1) member Cbx7 which promotes trimethylation of H3 at Lys-9 (H3K9me3)[30].

**Characterization of Cas9-induced mutation sizes and patterns**. In addition to mutation frequencies, we analyzed mutation sizes and patterns. We found that Cas9 targeting yielded mostly small mutations. On average, 73.4% of all deletions in the cell line were smaller than 10-bp (Fig. 4a). The most common sizes were {1,2}-bp with sgRNA2 and {2,4,5}-bp with sgRNA3, each accounting for approximately 12 to 14% of all deletions. In contrast, sgRNA1 led to a large number of 3-bp deletions (41.8%, Fig. 4a), the majority of which denoted the loss of triplet CGG, likely at positions 1–3 or 4–6 upstream of the PAM (40.2%, Fig. 4b). Single-nucleotide insertions accounted for > 78% of all insertions with sgRNA1/2, and > 95% with sgRNA3 (Fig. 4a). We found similar size and pattern distributions in experiments using varying Cas9 concentrations (Supplementary Figure 6).

We observed that deleted regions neighbored the expected Cas9 target site between nucleotides 3|4 upstream of the PAM with sgRNA1/2 (Fig. 4c). This indicates that resection might preferentially occur on one rather than both DNA ends at the break site. With sgRNA3, deletions often neighbored nucleotides 4|5 instead. In addition, each sgRNA led to predictable 1-bp insertions, namely G (63.7%) for sgRNA1, G (73.9%) for sgRNA2, and T (97.9%) for sgRNA3 (Fig. 4d). Deletion borders and 1-bp insertions were therefore highly consistent and guide-specific.

**Insight into Cas9 cleavage based on insertion patterns**. We sought to understand the observed preference for specific nucleotide insertions under current Cas9 cleavage models (Figs. 4d and 5a). Cas9 is thought to primarily induce blunt-ended DSBs[3,31]. Besides direct re-ligation, blunt ends may be processed leading to a deletion or, crucially, template-independent addition of nucleotides[9]. However, evidence from the seminal work on CRISPR-Cas9[3,31] and recent simulations on Cas9-domain conformation[32] suggests that Cas9 may also generate staggered DSBs. Specifically, Cas9 domains RuvC and HNH

could cleave between nucleotides 3|4 on the target DNA (tDNA) and 4|5 on the non-target DNA (ntDNA) upstream of the PAM[32]. The resulting 5′ overhangs could trigger polymerase-based fill-in at position 4, producing sgRNA-specific insertions consistent with our data (Figs. 4d and 5a).

**Plausible Cas9 DNA cleavage and repair models**. We assessed three different models of Cas9 cleavage and DNA repair (Fig. 5a). First, a blunt model producing blunt-ended DSBs primarily at 3|4[3,31], and leading to insertions of a random nucleotide[9]. Second, a staggered model cleaving at 3|4 (tDNA) and 4|5 (ntDNA)[32] and inducing the replication of nucleotide 4. Third, a combination of both blunt and staggered models. Each model determined an expected distribution of insertion counts across sites in the target sequence. In order to compare model distributions against observed insertions, we also addressed uncertainties caused by addition of nucleotides identical to neighbors in the target sequence (e.g. ambiguity in insertions A**A** and **A**A). We achieved this by redistributing the counts of ambiguous insertions based on the counts of unambiguous insertions and rules derived from the models (see Methods, Fig. 5b). Figure 5b shows observed counts of unambiguous insertions (filled bars) and model-corrected counts of ambiguous insertions (unfilled bars) obtained for a range of possible break sites on the non-target DNA in our cell line Cas9 assays using sgRNA1-3. Each row depicts the redistribution of ambiguous counts for each assay according to a different model, and the vertical shaded areas highlight the main break sites of interest located 3|4 (right) and 4|5 (left) nucleotides upstream of the PAM. For each plot, we also include an insert labelled "expected", with an illustration of the expected model-based count distribution for the two sites 3|4 and 4|5.

**Blunt DNA cleavage-repair model**. Based on the blunt model we expected to see template-independent insertions, and therefore similar insertion frequencies for every nucleotide at the break site (equal-height bars for sites 3|4 and 4|5 in "expected" inserts, top row plots of Fig. 5b). When we redistributed the observed ambiguous counts in a nucleotide-unbiased manner, following the frequency of unambiguous insertions at each site, we did not obtain the expected nucleotide-unbiased insertion profiles. Specifically, redistributed counts showed significantly more insertions of G (sgRNA1), G (sgRNA2), and T (sgRNA3) at sites 3|4 and 4|5 (top row in Fig. 5b, note the log-transformed scale). We further note that the nucleotide imbalance would be unavoidable for the most targeted sites, 3|4 and 4|5, given the substantial frequency gap observed between the most inserted and the remaining nucleotides.

**Staggered DNA cleavage-repair model**. Alternatively, the staggered model determined template-dependent insertions following a 1-nt 5′ overhang on the opposite strand. According to this preference, we expected to see significantly more insertions of the nucleotide identical to the DNA base immediately downstream of the break site ("expected" inserts in middle row plots of Fig. 5b). For the staggered model, we redistributed ambiguous insertions in our observed counts based on both the frequency of unambiguous insertions, and the 1-nt 5′ overhang template. The results obtained for this model showed the expected template-based insertion pattern at break site 4|5 for all sgRNAs ("expected" inserts *vs.* main plots in middle row of Fig. 5b). However, the results did not follow the expected pattern at break site 3|4 for sgRNA1 and sgRNA3. Specifically, results showed similar frequencies of the four nucleotides, typical of template-independent insertions in a blunt model, while nucleotides C (sgRNA1) and A (sgRNA3) should have been significantly more frequent than

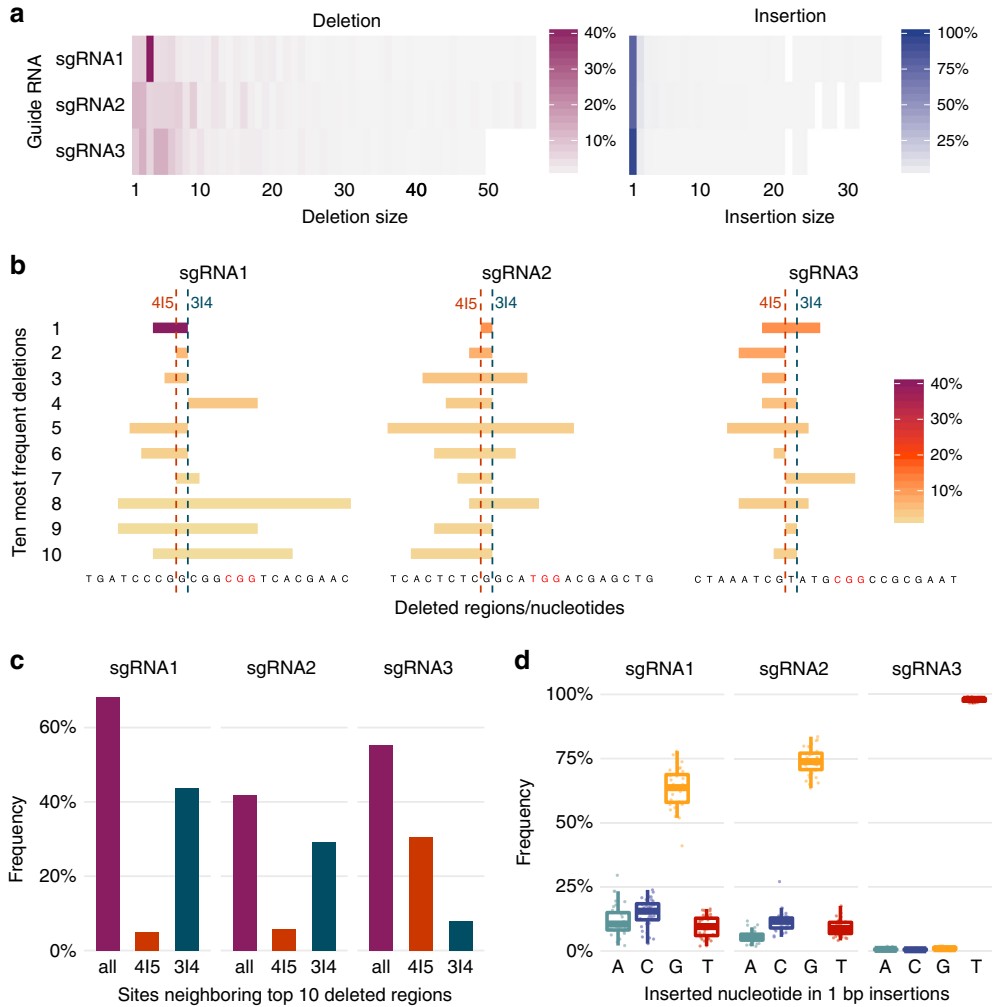

**Fig. 4** Mutation patterns induced by Cas9 in the 36-integration TRIP cell line. **a** Observed deletion and insertion sizes. Heatmaps show the overall frequency (color gradient) of deletions (red, left) and insertions (blue, right) per size (horizontal axis) for each guide RNA (vertical axis) in the TRIP cell line. **b** Deletion patterns and positions. Shown for each guide RNA are the ten most frequent deletion patterns with respect to the non-target DNA, from top to bottom in decreasing order of frequency. Each horizontal bar indicates the position of a deletion pattern, and corresponding non-target DNA sequence lost (at the bottom), colored according to frequency. Expected 3|4 and alternative 4|5 break sites are indicated by two vertical dashed lines. **c** Frequency of sites neighboring the ten most frequently deleted regions for each guide RNA, shown in Fig. 3b. Three vertical bars indicate the proportion of: all such deletions regardless of neighboring site (all, red), the subset of those deletions neighboring the expected break site (3|4, green), or the subset of those deletions neighboring the alternative break site (4|5, orange). For deletions with ambiguous positions, we weighted the frequencies by the ratio of positions meeting the criteria. We observed similar trends using all data. **d** Frequency of each nucleotide in 1-bp insertions. For each guide RNA, boxplots show the frequency (vertical axis) of insertions of each nucleotide (horizontal axis and color) across the 36 IRs (dots). Boxes show the median, first and third quartiles of the frequency distributions; whiskers extend to 1.5 times the inter-quartile range from the top and bottom of the box. Source data are provided in the Source Data file

others based on the staggered model (main plots *vs.* "expected" inserts in middle row of Fig. 5b).

**Combined blunt and staggered DNA cleavage-repair model**. Finally, the combined blunt and staggered model provided the best fit: (i) most insertions occurred at sites 3|4 and 4|5, the expected primary targets of blunt and staggered cleavage on the non-target DNA (bottom row of Fig. 5b); (ii) the four nucleotides were similarly likely at site 3|4, consistent with template-independent insertions at blunt DNA ends; and (iii) the most inserted nucleotide at site 4|5 matched the downstream neighbor, as expected upon fill-in of 1-nt 5′ overhangs. A combination of the blunt model with an alternative staggered model inducing 1-nt 3′ overhangs at break site 3|4 (ntDNA) could possibly fit as well, although we found no reference to such a model in the

literature. We note that our data could include re-cleavage events leading to a biased selection for mutations. Specifically, accurately ligated DNA ends can be re-cleaved, while mutations become nearly fixed in the population, since Cas9 is less likely to recognize and cleave at a mutated site. Over time, this could skew the ratio between mutated and wild-type sequences, and influence blunt vs. staggered patterns. For this reason, we limited the scope of our analysis to showing that both patterns occur, without quantifying how frequent each one is.

**Staggered DNA ends in Cas9-targeted mouse pre-B cells**. We investigated whether 1-nt 5′ overhangs would be generated by Cas9. For this purpose, we collected independent data generated by hairpin capture and sequencing of DNA end structures at Cas9 DSB sites (HCoDES)[33]. Specifically, we re-analyzed DNA ends

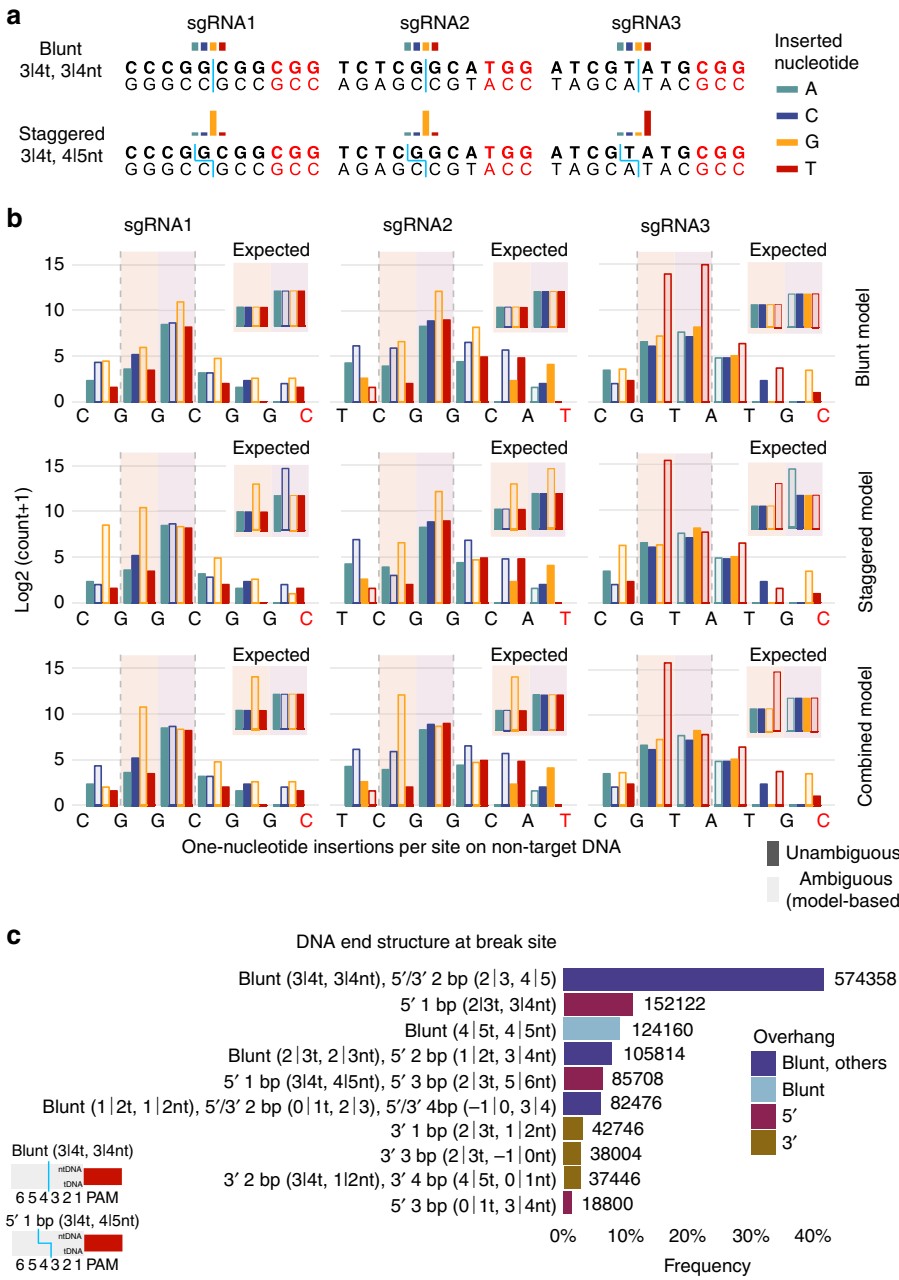

**Fig. 5** One-nucleotide insertion patterns and DNA end structures at the break site. **a** Illustration of blunt and staggered DNA ends at the break site, and expected distribution of 1-bp insertions of the four nucleotides following DNA repair. Double-stranded sequences including PAM and 8-bp upstream, with bottom and top denoting target and non-target DNA. Blue straight and staggered lines through the sequences indicate blunt and staggered DNA ends. Colored bars on top sketch the expected distribution of 1-bp insertions upon DNA repair. Blunt model: blunt-ends primarily at 3|4 upstream of the PAM (straight line), resulting in template-independent insertion and thus similar frequencies of the four nucleotides (uniform distribution, similar-height colored bars). Staggered model: staggered ends mostly with termini at 3|4 (tDNA) and 4|5 (ntDNA) upstream of the PAM (staggered line), with template-dependent fill-in resulting in a skewed distribution with most insertions of the DNA base identical to nucleotide 4 (unequal-height colored bars). **b** Unambiguous insertion counts (filled bars) and ambiguous insertion counts (empty bars) redistributed according to blunt, staggered, and combined models. Shown are insertion counts (vertical axis) of each nucleotide (color) per site on the ntDNA (horizontal axis). Vertical shaded areas indicate the 3|4 and 4|5 sites upstream of the PAM. Unambiguous counts are directly determined from the data (filled bars), whereas ambiguous counts are redistributed over windows of ambiguous sites (empty bars) based on: (i) relative proportions of unambiguous counts, and (ii) likelihood of each nucleotide insertion according to the cleavage model. (**c**) Re-analysis of DNA ends generated by Cas9 targeting of a region on chromosome 6 in mouse pre-B cells deficient in DNA Ligase IV and arrested in G1 phase. Bar length denotes relative frequency, shown for the ten most frequent DNA end structures accounting for ~91% of all unique patterns in the data. Absolute frequencies are displayed. Multiple DNA end structures associated with the same sequence are grouped with a single bar and label. Bars are colored by type of structure. The bottom left figure shows an illustration of two DNA structures: blunt (3|4t, 3|4nt), and 5′ 1-bp overhang (3|4t, 4|5nt). Source data are provided in the Source Data file

induced by Cas9 targeting to a region on chromosome 6 in mouse pre-B cells arrested in G1-phase and LigaseIV-deficient[33]. Many DNA end structures could not be uniquely mapped (Fig. 5c). We saw prevalent blunt ends at 3|4, although these could not be discerned from certain 2-nt 5′/3′ overhangs. Specifically, blunt ends at 3|4 accounted for ~20% when aligning sequences to the reference as-is, as performed in the original study[33]. We also determined a proportion of ~41.6% upon masking the noise caused by incomplete bisulfite conversions. Shorter 1-nt overhangs were favored. Additionally, we saw a preference for 5′ overhangs, particularly 1-nt 5′ at 2|3 (tDNA) 3|4 (ntDNA) accounting for ~11.0%, and at 3|4 (tDNA) 4|5 (ntDNA) confounded with a 3-nt 5′ overhang (~6.2%).

## Discussion

As the CRISPR-Cas9 system is widely used for gene editing, understanding Cas9 activity across the genome is crucial to identify cleavage and mutation patterns enabling new applications or improvements. Using a combination of the CRISPR-Cas9 and TRIP technologies with high-throughput DNA sequencing, we characterized mutations at ~1k loci throughout the genomes of mES cells. We showed high reproducibility in a single-promoter TRIP cell line with 36 IRs and pools of cells with thousands of heterogeneous multi-promoter TRIP IRs.

In line with previous studies[13], we saw sgRNA-dependent variation in Cas9-induced mutation frequency. Specifically, sgRNA3 led to larger mutation frequencies than sgRNA1/2. We reasoned that the high GC-content of sgRNA1/2 (75%, 70%) relative to sgRNA3 (45%) could explain the lower efficiency of sgRNA1/2, based on reports associating low and high GC-content with reduced Cas9 activity[12]. Overall, sgRNA1/2 produced comparable mutation frequencies, possibly given the high similarity including a 19-bp reverse complement overlap. Genomic location was the largest contributor to variation in mutation frequency. However, mutation frequency correlated weakly with TGE features of wild-type cells and reporter expression in TRIP cells. Importantly, reporter expression and mutation frequency were quantified in TRIP mES cells bearing identical integrations and epigenetic landscape. This result suggests that the effects of genomic context on mutation frequency may be influenced by other factors than those surveyed here. Although studies using doxycycline-controlled chromatin states on engineered targets previously showed effects on mutation efficiency[19,21], an approach targeting endogenous sites in human cells revealed only modest effects with effect sizes that were largely dependent on guide RNA and possibly other factors[22]. We saw similar results in our study, with generally higher correlations between TGE features and mutation frequency using sgRNA3. We note that, due to the exogenous manipulation of chromatin states, those studies could produce larger contrasts between permissive and non-permissive states than those seen in the context of our work. Our findings were also consistent with two other studies assessing the association between expression or epigenome and mutation frequency at endogenous target sites[17,24]. One study reported weak correlations overall across developmental stages in zebrafish, which were slightly higher for expression than epigenetics[24]. The other study reported that native chromatin accessibility and DNA methylation were predictive of Cas9 binding, but Cas9 binding was uncorrelated with mutation frequency in mES cells[17]. Together, these findings suggest that genomic context likely influences Cas9 binding and cleavage, but that the association with mutation frequency can be further modulated by additional factors such as guide RNA sequence. It is possible that the correlation could be disrupted by stochasticity in the outcome of DNA repair, especially as a result of potential re-cleavage events. We also reason that dynamic reorganization of the regulatory landscape

during the S-phase of the cell cycle could influence estimates of Cas9-induced mutation frequency but not IR expression, which is mainly produced in G1-phase. In particular, mES cells are known to exhibit unusually short G1 and long S-phases[34]. In S-phase, the DNA is unpacked to allow for replication, enabling unperturbed Cas9 binding to otherwise inaccessible loci. This could mean that most Cas9 cleavage in our assays occurred in S-phase under widespread DNA accessibility, masking differences between permissive and non-permissive domains, and hence lowering the association of mutation frequency with TGE features. Nevertheless, insertions were more dependent on the regulatory landscape than deletions, particularly concerning influencers of transcriptional activity genome-wide such as PolII complex, histone marks or Lamin-B1. We noted that deletions varied in size and pattern, while insertions comprised mostly sgRNA-specific single-nucleotides. Specifically, deletions could be generated by a variety of DNA processing events on blunt or staggered DNA ends, either aimed at ligation by NHEJ throughout the cell cycle[35] or generation of 3′ overhangs in preparation for HDR in S and G2 phases[36]. In contrast, insertions likely arose by gap-filling of specific overhangs by a DNA polymerase. We hypothesize that the deterministic nature of insertions could preserve the association with TGE features better than the range of processes involved in deletions, possibly yielding varied dependencies on the regulatory landscape.

We observed small Cas9-induced lesions with all sgRNAs. The most common deletion was loss of trinucleotide CGG, accounting for ~40% of all deletions with sgRNA1. We attributed this event to microhomology-mediated end-joining (MMEJ), which could rely on the repetition of CGG in sgRNA1 to align the DNA ends resulting in the loss of triplet CGG[37]. Regions deleted with sgRNA3 neighbored nucleotides 4|5 (ntDNA) rather than the expected target site 3|4, seen with sgRNA1/2, suggesting alternative Cas9 targeting between 4|5 (ntDNA). In addition, single-nucleotide insertions showed high prevalence of a specific nucleotide per sgRNA, consistent with independent studies targeting the HPRT, AAVS1 and TREX1 genes in HCT116 cells[38], the GFP, NDC1 and LBR genes in K562 cells[39], as well as numerous other regions in HCT116, HEK293, and K562 human cells[38–40]. We determined that the frequently inserted DNA base matched nucleotide 4 upstream of the PAM, which hinted at eventual template-dependent repair of staggered DNA ends with termini at 3|4 and 4|5 on opposite strands. Different Cas9 cleavage and DNA repair models were analyzed, seeking to explain observed insertion patterns[3,31,32]. We propose a Cas9 cleavage model inducing primarily blunt and occasionally staggered DNA ends (Fig. 6). Most blunt ends are ligated, possibly upon resection, resulting in either wild-type or deletion. On occasion, blunt ends may lead to an insertion via template-independent addition of a random nucleotide (e.g. by Pol μ)[9]. We showed that insertions mainly derived from DNA ends with 1-nt 5′ overhangs, specifically with termini at positions 3|4 (tDNA) and 4|5 (ntDNA)[32]. We noted that 1-nt 3′ overhangs could also explain the observed insertions. However, re-analysis of independent data on DNA end structures generated by Cas9 targeting in mouse DNA Ligase IV-deficient pre-B cells[33], revealed prevalence of blunt ends and preference for 1-nt 5′ over 3′ overhangs, in accordance with our hypothesis. We cannot exclude the possibility that the overhangs would arise by minimal resection of blunt ends rather than directly by Cas9 cleavage. Regardless of the process, we presented evidence that staggered DNA ends are generated and likely responsible for most insertions. The insertion patterns we observed in mES cells mostly in S-phase were corroborated by independent Cas9 targeting in human cells HEK293, HCT116 and K562[38–40], as well as in mouse pre-B cells arrested in G1-phase[33], using different plasmid systems. The consistency of

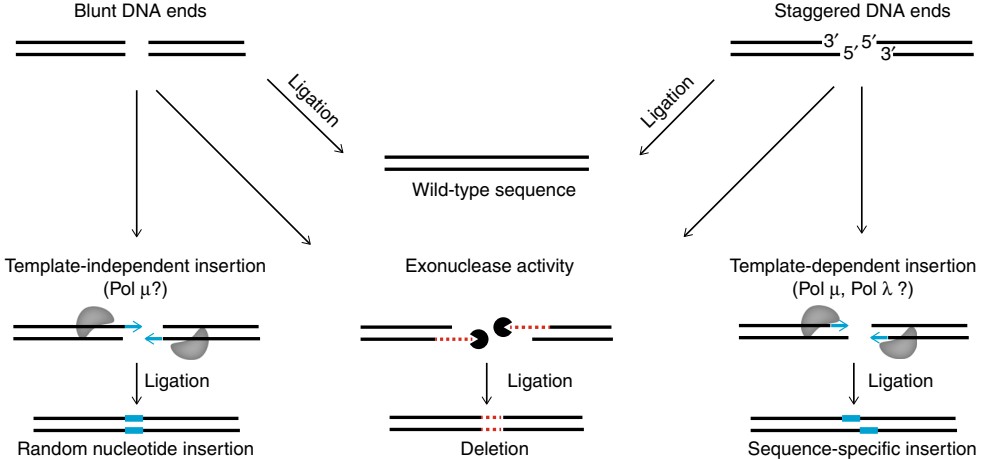

**Fig. 6** Illustration of DNA repair outcomes after Cas9-induced double-strand break. Both blunt and staggered ends can be directly ligated back into wild-type sequence, or generate a deletion through resection by nuclease activity prior to ligation. Blunt ends can also result in an insertion by template-independent addition of a random nucleotide, possibly established by Pol μ. Staggered ends lead primarily to template-dependent insertions, possibly established by polymerases such as Pol μ or Pol λ

these results indicates that our findings might generalize beyond the scope of our assays.

Here, for the first time, Cas9-induced mutation patterns were characterized at thousands of target sequences embedded throughout the genomes of mES cells. We laid out the likely combination of mechanisms of Cas9 cleavage and DNA repair underlying predictable 1-nt insertion patterns reported in the literature[38–40]. In particular, we revealed that Cas9 occasionally generates breaks with staggered DNA ends. These could be leveraged to increase knock-in efficiency and control the orientation of inserts into host DNA by homology dependent or independent insertion[41]. Furthermore, our data indicated that guide RNA sequence determines the frequency of staggered ends, ultimately influencing insertion and overall mutation efficiency. This result suggests that guide RNAs could be designed to maximize insertion frequency. Extensive testing of a wider range of guides will be needed to build models that predict insertion efficiency based on relevant guide RNA features. Our findings could have important implications for the optimization of Cas9-mediated knock-in, which remains a major challenge in genome editing.

In addition, we introduced the TRIP technology as a tool to multiplex RNA-guided Cas9 targeting to regions in reporter genes integrated genome-wide. This CRISPR-on-TRIP approach allowed us to seamlessly isolate target sequence from genomic location, and perform sequence-independent analysis of variation in Cas9 mutation efficiency and patterns in different genomic contexts. We found that genomic location is a key determinant of mutation frequency, which however correlated modestly with genomic and epigenomic context. We reasoned that guide RNA and stochasticity in the outcome of DNA repair, together with potential re-cleavage events and the reprogramming of the epigenetic landscape during the cell cycle could contribute to this result. Further investigation will be needed to clarify the impact of genomic context. Finally, we demonstrated that CRISPR-on-TRIP is a promising tool to profile Cas9 activity at a large number of target sequences scattered throughout the genome, and can be combined with other assays to study the influence of a variety of processes on Cas9 activity and induced mutation patterns (Fig. 7).

## Methods

**Construction of TRIP plasmid libraries**. The monoclonal TRIP cell line used in this work was established in our previous TRIP study[25]. The piggyBac-based template vector pPTK-Gal4-mPGK-Puro-IRES-eGFP-sNRP-pA carrying the

reporter unit in the TRIP cell line library comprised the following elements: piggyBac 5′-TR, 14 Gal4 binding sites, mPGK promoter, puromycin resistance (PuroR) coding sequence, encephalomyocarditis virus internal ribosome entry site (IRES), EGFP coding sequence, PstI site (used to clone barcodes) + DpnII site (used to map IRs), human soluble neuropilin-1 (sNRP-1) polyA signal, and piggyBac 3′-TR (67-bp). Barcoded inserts were generated through amplification of 5 ng template vector pPTK-Gal4-mPGK-Puro-IRES-eGFP-sNRP-pA (GenBank KC710227), using primers PB-barcode-long-7 (5′-GTGACACCTGCAGGATCA (N)₁₆CTCGAGTTGTGGCCGGCCCTTGTGACTG-3′, where (N)₁₆ denotes a random 16-nt long reporter barcode) and PB-barcode-short-7 (5′-GACATA ACGCGTATACTAGATTAACCCT-3′). After PCR purification, the PCR product was digested with restriction enzymes PstI and MluI (underlined). In parallel, the pPTK-Gal4-mPGK-Puro-IRES-eGFP-sNRP-pA vector was digested with the same restriction enzymes and then dephosphorylated. The digested PCR product was next ligated with the prepared vector using 10U of T4 DNA ligase (Promega). The resulting ligation product was transformed into electrocompetent *E.coli* cells and the plasmid DNA (TRIP plasmid library) was isolated using Genopure plasmid maxi kit (Roche).

For the multi-promoter TRIP pool established in this study we first generated seven libraries, each containing reporter constructs with one of seven different promoters: CMV, cMyc, Hoxb1, Nanog, Oct4, p53 and PGK. These were included to assess effects of promoters with different characteristics, such as strength[42], exogeneity, housekeeping status, retinoic acid-inducibility[43–46], and TATA-less status[45,47–50]. The piggyBac-based plasmid vectors carrying the reporter unit comprised the following elements: piggyBac 5′-TR (314 bp long), promoter of interest, EGFP coding sequence, DpnII site (used to map IRs), 5-bp promoter index unique to the promoter, KpnI site (used to clone barcodes), sNRP-1 polyA signal and piggyBac 3′-TR (242-bp). Barcoded inserts were generated by amplification of 5 ng PB template vector using primers Kpn-RandomBC-1 (5′-AAAAGGTACC (N)₁₈GAGTTGTGGCCGGCCCTTGTGACTG-3′, with (N)₁₈ denoting a random 18-nt long reporter barcode) and BssH2-A (5′-AAAAGCGCGCATACTAGAT TAACCCTAGAAAGATAATCATATTG-3′). After PCR purification, the barcoded inserts were digested with restriction enzymes KpnI and BssHII (underlined). In parallel, the plasmid vectors were digested with restriction enzymes KpnI and MluI (the latter generates sticky ends compatible with those made by BssHII) and subsequently dephosphorylated. Ligation of the digested barcode insert into the digested plasmid vectors and electrotransformation of bacterial cells were performed as described above. The seven promoter-specific plasmid libraries were mixed together in the following molar ratios: 2:2:4:2:2:2:1 to obtain the multi-promoter TRIP library.

**Cell culture and TRIP library transfection**. Mouse embryonic stem (mES) cells EBRTcH3 expressing the tetracycline-controlled transactivator (tTA) from the endogenous ROSA26 promoter (EStTA)[51] were cultured in 60% BRL cell-conditioned medium in the presence of 10% fetal calf serum (FCS), leukemia inhibitory factor, MEK inhibitor PD0325901, and GSK-3 inhibitor CHIR99021[52]. The EBRTcH3 ES cells were provided by the te Riele lab (Netherlands Cancer Institute), which had received them from Dr. Masui (International Research Center of Japan). The EBRTcH3 ES cells were originally derived from E14tg2a ES cells[53] by Masui and colleagues[51]. Culture dishes were coated with 0.15% gelatin and incubated at 37˚C for one hour before plating.

For the cell line, six million mES cells were plated and incubated for 4 h at 37˚C. The cells were then transfected with 22.5 μg of the mPGK TRIP plasmid library

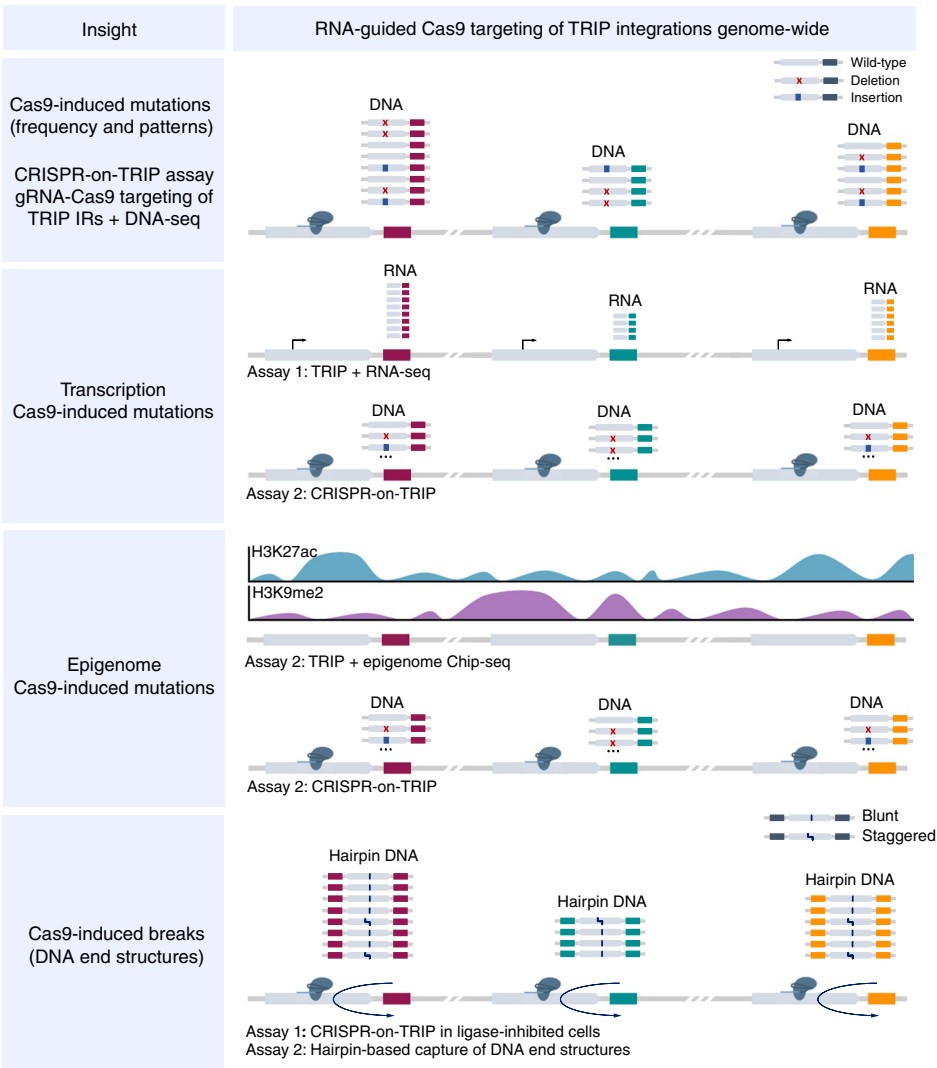

**Fig. 7** Potential applications of CRISPR-on-TRIP. RNA-guided Cas9 targeting of regions within integrated TRIP reporters (CRISPR-on-TRIP) can be combined with other assays to investigate effects of various processes on Cas9-induced mutation frequency and patterns

together with 2.5 μg of mouse codon-optimized version of PB transposase (mPB) plasmid[54] using Lipofectamine 2000 (Invitrogen) and incubated at 37 ˚C. After 48 h of incubation, the transfected cells were sorted through fluorescence-activated cell sorting (FACS), selecting single cells with "medium" levels of EGFP expression, which were used to establish stable TRIP cell lines. The cell line EStTA-PB-B-18 used in this study carries 36 IRs.

For the TRIP pool, nine million mES cells were plated and incubated for four hours at 37 ˚C. The cells were then transfected with 15 μg of the multi-promoter TRIP plasmid library mixture and 5 μg of PB transposase plasmid mPB-L3-ERT2. TatRRR-mCherry plasmid using Lipofectamine 2000 (Invitrogen) and incubated at 37 ˚C. After 24 h, the cells were FACS-sorted to select for a desired mCherry signal from the PB transposase plasmid[26]. The sorted cells were resuspended in mES medium with 1 μM of tamoxifen to activate the PB transposase. After 24 h, the cells were washed and resuspended in mES medium without tamoxifen, in which they were cultured for 5 days, refreshing the medium every other day. Biological replicate pools were established by subculturing several aliquots of the pool, which were grown for another week. Technical replicates were established by splitting each pool into two and growing each half separately for another week. The multi-promoter TRIP pool used in this study was grown from approximately 500 cells containing an average of ~ 25 IRs per cell.

**Determination of genomic location and expression of IRs**. Barcoded IR regions were extracted by inverse PCR followed by high-throughput DNA sequencing. The resulting reads were subject to quality control, including the filtering of aberrant barcodes arising from errors induced by PCR or sequencing. After pre-processing, the regions were aligned against the mouse genome assembly mm9 in order to map IR loci. Expression of IRs was determined by sequencing of reverse transcribed RNA (cDNA). Barcode abundance was also estimated by sequencing genomic

DNA (gDNA), and used to normalize IR expression. These procedures were performed according to the TRIP protocol[26].

**Association of TRIP integrations with genome-wide features**. In addition to IR expression, we assessed endogenous gene expression[25] and a range of regulatory elements measured genome-wide in mES cells. These data were previously collected from public sources and preprocessed aiming to maximize comparability[28]. Most features were obtained from ChIP-seq data on epigenetic modifications such as DNase I hypersensitivity, histone marks, and binding of transcriptional and epigenetic regulators. For ChIP-based features, a score was defined as the log2-transformed ratio between signal and control. Genomic features including GC content and gene proximity, among others, were derived from the mouse reference genome mm9 and Ensembl genes (release 66). Specifically, proximity measures were defined as the negative log2-transformed distance ( +1) to the nearest concerned genomic feature (e.g. gene, transcription start site). Chromatin compaction was estimated from Hi-C data as the rate of decay in contact probability $\alpha$ between two loci with increasing genomic distance, locally approximated in 400 kb windows by a power-law function with scaling exponent $\alpha$. Finally, the association between every IR and feature pair was computed as the mean normalized score of the genomic or regulatory feature over a region of 2 kb surrounding the integration site.

**sgRNA and ssODN design**. All oligonucleotides used in this study were purchased from Integrated DNA Technologies. We designed three sgRNAs targeting different regions within the EGFP reporter gene, using the CRISPR design tool http://crispr.mit.edu/ (Supplementary Table 3). We considered three primary criteria. First, purposed sgRNA-targeting of a sequence within the EGFP gene body in close proximity to the barcode, enabling reliable amplification of both barcode and target

site. Second, reliability according to the CRISPR design tool, with no or minimal reported off-target sites. Third, EGFP recognition in both sense and anti-sense orientation, to exclude orientation-dependent effects. In addition, we designed a 141-bp single-stranded oligodeoxyribonucleotide (ssODN) template for HDR-based knock-in in with sgRNA2. Our ssODN comprised a 21-bp sequence for knock-in with ~ 60-bp homology arms at each side. Following the recommendations in https://www.addgene.org/crispr/zhang/faq/ (Supplementary Table 3), we avoided overlap between the ssODN homology arms and the barcode located downstream of the EGFP reporter gene. We also designed the junction of the homology arms < 10 bp from the Cas9 target site. Lastly, we designed the ssODN template with 58 and 62 nucleotide-long homology arms, in agreement with the recommended 50–80 range.

**Cloning and transfection of sgRNA-guided CRISPR/Cas9**. We used human codon-optimized SpCas9 and chimeric guide RNA expression plasmid pX330-U6-Chimeric_BB-CBh-hSpCas9 to complex Cas9 with different sgRNAs (1–3). In addition, we used mCherry as a fluorescent marker for visualization and sorting of Cas9-sgRNA transfected mES cells. All Cas9 targeting assays experiments were performed in triplicate. For cell line experiments, we co-transfected the Cas9-sgRNAs and mPB-L3-ERT2.TatRRR-mCherry plasmids. Ten million mES cells were first seeded on a 10-cm dish. Four hours later, the cells were transfected with 13.5 µg of Cas9-sgRNA and 1.5 µg of mPB-L3-ERT2.TatRRR-mCherry plasmids using 45 µl of Lipofectamin 2000 (Invitrogen). For TRIP pool experiments, we cloned mCherry-expressing Cas9-sgRNA plasmids. We first digested the Cas9-sgRNA plasmids with 10 units NotI (Roche) and 10 units SbfI (NEB). We further PCR-amplified CMV-driven tatRRR-mCherry from mPB-L3-ERT2.TatRRR-mCherry using primers Fragment.FOR (10 µM) and Fragment.REV (10 µM) (Supplementary Table 3). The digested Cas9-sgRNA plasmid and PCR-amplified tatRRR-mCherry fragment were then cloned using Gibson Assembly Master Mix (NEB). To avoid reduced transfection efficiency of mCherry-expressing Cas9-sgRNA plasmid and ssODN with Lipofectamin 2000, we followed the protocol from Nucleofector™ Kit for Mouse Embryonic Stem Cells (Lonza) instead. Five million mES cells per condition were trypsinized, spun down and resuspended in 90 µl of Mouse ES Cell Nucleofector™ solution. They were thereafter transfected with a total of 6.25 µg of DNA (3.75 µg mCherry-expressing Cas9-sgRNA and 2.5 µg ssODN) using the program A-024 of the Nucleofector™ Kit for Mouse Embryonic Stem Cells (Lonza). Transfected cells were resuspended in 500 µl of pre-warmed culture medium and plated in gelatin-coated 10 cm dishes. Cells were sorted 24 h after transfection. For disruption assays, we transfected five million mES cells with 8 µg of mCherry-expressing Cas9-sgRNA. For editing, five million cells were transfected with 4 µg of mCherry-expressing Cas9-sgRNA together with 1 µg of designed ssODN containing the 21-bp sequence for knock-in with ~ 60-bp homology arms. We decided to use sgRNA2 in editing assays, given that the short distance between the region targeted by sgRNA and the IR barcode prevented the design of proper ssODN homology arms (Supplementary Figure 8). Additionally, sgRNA2 was preferred over sgRNA1 due to its larger mutation efficiency. In all experiments, transfected mCherry-positive mES cells were sorted by flow cytometry (MoFlo), collected in conditioned media containing 20% FCS, and subsequently spun down plated in conditioned media containing 10% FCS. Finally, Cas9-sgRNA-transfected mES cells were expanded for five days before isolating genomic DNA.

**DNA isolation and preparation of samples for sequencing**. After incubation of Cas9-sgRNA-transfected mES cells, DNA was isolated using DNeasy Blood & Tissue Kit (Qiagen) and prepared for sequencing. We PCR amplified cell line DNA IR regions of 414 bp surrounding the sgRNA target sites using 10 µM of PB-cDNA-forward-1-BC primers, containing different index sequences for multiple reactions, and 10 µM of PB-cDNA-Reverse-5 primer (Supplementary Table 4). The PCR product was sent for Illumina MiSeq. In TRIP pool experiments, we tagged amplicons using 16-nucleotide Unique IDentifiers (UIDs) to be able to detect multiple readings of the same DNA molecules (Supplementary Table 5)[55]. We performed all Cas9 targeting assays in triplicate. Barcoded IR regions were PCR amplified using Phusion® High-Fidelity DNA Polymerase (NEB) following the protocol in Supplementary Table 6. We used Exonuclease-I (20 units) (Enzymatics) for cleavage of single stranded DNA. Purified sequences were sent for Illumina HiSeq. Replicates were assessed for consistency of IR mutation frequencies, and merged for subsequent analyses.

**Validation of promoter-barcode index association**. For identification purposes, each promoter was originally associated with a unique 5-nucleotide index located 44 base-pairs downstream of the IR (Supplementary Table 2). We performed Sanger sequencing on the DNA extracted for five randomly selected IRs in the TRIP pool to confirm the absence of recombination and the correct association between barcodes and corresponding promoters. We designed primers for nested PCR, including two IR specific primers, PB-Valid.3-Out primer-1 and Inner-1, and locus-specific genome spanning primers, GEMP_mvla_outer and GEMP_mva-l_inner (Supplementary Table 7). PCR amplification was performed in two steps using the primer combinations PB-Valid.3-Out primer-1/GEMP_mval_outer for PCR1 and Inner-1/GEMP_mval_inner for PCR2. The resulting products were diluted in water (1:5) and prepared for Sanger sequencing using 10 µM of primers together with Big Dye terminator version 3.1.

**Identification and characterization of Cas9-induced lesions**. We parsed the DNA reads obtained from Cas9-targeted TRIP cells in order to map barcodes and UIDs[25], and extract the sequences of interest containing the 20-nucleotide region targeted by Cas9. These were aligned against the wild-type sequence to identify and characterize Cas9-induced lesions. We used semi-global alignment with the following weights: match + 2, mismatch −2, gap opening penalty −5, gap extension penalty −0.5, and initial score 30. For TRIP cell line experiments with sgRNA1, sgRNA2 and sgRNA3, respectively 96%, 95% and 91% of the reads could be parsed, whereas TRIP pools yielded between 86% and 89% of successfully parsed reads. Quality control and filtering of aberrant barcodes were further performed as described in previous work[25].

**Analysis of mutation frequencies, sizes and patterns**. We analyzed sequenced IR regions with a read coverage of at least 30 in all Cas9 assays on TRIP cells. The average read coverage per IR was significantly higher in the TRIP cell line than the pool (Supplementary Figure 9). However, the cell line contained only 36 barcoded regions whereas the pool offered ~ 1k IRs. As a result, we used TRIP pool data primarily to assess overall trends of variation in IR mutation frequency throughout the genome, and high-resolution TRIP cell line data to identify mutation sizes and patterns.

Relative frequencies of each lesion type were calculated per IR as the ratio between the number of reads exhibiting such lesion and all reads for the given IR (Figs. 2a–c). We plotted frequencies per lesion type for each of 36 IRs in the TRIP cell line (Fig. 2a), as well as distributions (Fig. 2b) and stacked medians stratified by promoter (Fig. 2c) considering all 1359 IRs shared by TRIP pool assays. To assess variation in IR mutation frequency with genomic locus in the TRIP pool, we fitted linear regression models describing the relationship between mutation frequencies in each pair of assays and determined corresponding $R^2$ and F-statistic $p$-values (Fig. 2d). Additionally, we fitted a global linear regression model to IR mutation frequencies in all TRIP pool assays taking into account genomic locus, guide RNA, knock-in, promoter and a significant interaction term between genomic locus and guide RNA (Fig. 2f). Based on this model, we determined $\eta$-squared values denoting the effect size or variance explained by each independent variable (Fig. 2g). For this, we relied on multi-way ANOVA tests with partial sum of squares preserving the principle of marginality (type II sum of squares). The ratio between knock-in and other insertions was calculated per IR and plotted against binned total IR mutation frequency together with the mean and confidence interval of 0.95 obtained by nonparametric bootstrapping (Fig. 2d). Association between IR total mutation frequency and regulatory element scores was determined using Pearson's correlation, along with the corresponding t-test $p$-values corrected for multiple testing using the Benjamini-Hochberg procedure (Figs. 3b–d). We calculated and plotted overall deletion and insertion frequencies per size in the TRIP cell line (Fig. 4a). In addition, we determined the frequencies of individual deletion patterns (Fig. 4b) and the distributions of insertions of the four DNA bases in predominant one-nucleotide insertions (Fig. 4d). Deletions with ambiguous positions could not be distinguished. We took this ambiguity into account when calculating the frequency of deletions neighboring the expected or alternative break site. Specifically, we weighed down the frequency according to the ratio of ambiguous positions neighboring the corresponding sites (Fig. 4c). All differences between distributions reported in the manuscript were determined using nonparametric two-tailed Wilcoxon tests. All given confidence intervals are calculated for confidence level 0.95.

**Analysis of one-nucleotide insertion patterns**. In order to assess different models[3,31,32] of Cas9 cleavage, we had to first resolve ambiguity in insertion sites. Ambiguity arises due to insertions of DNA bases next to matching nucleotides in the target sequence. For instance, insertions of an A immediately up or downstream of a wild-type A cannot be distinguished, as they result in identical mutated sequences (**A**A and A**A**). Since these results are confounded, both positions represent ambiguous insertion sites for the added base A. These observations are also true for longer and more complex sequences. Additionally, the deterministic nature of variant callers means that an insertion is called where the change is identified, thus once the repetition is detected. When processing sequences 5′ to 3′, insertions above would always be identified as occurring downstream of the wild-type A (A**A**). We redistributed accumulated insertion counts throughout ambiguous sites based on unambiguous insertions and the Cas9 cleavage and DNA repair models as follows (Fig. 5b). The wild-type non-target sequence was processed to identify ambiguous DNA base(s) at each site, essentially those matching the 5′ and 3′ neighbors in the wild-type sequence. Individual ambiguous sites were extended to maximal windows of consecutive ambiguous sites for the same DNA base. We performed the redistribution of ambiguous DNA base insertions within these windows. For blunt ends with template-independent insertions, without bias towards specific DNA bases, ambiguous counts were distributed to follow the relative proportions of unambiguous insertions at the sites within the window. For staggered ends repaired by template-based insertions, we classified sites within a window as likely or unlikely based on the expected insertion pattern. Considering a

1-bp 5′ overhang model, ambiguous sites with downstream nucleotide identical to the inserted DNA base would be considered likely whereas those with a different downstream nucleotide would be unlikely given the conformation of the DNA ends. We re-calculated the counts for ambiguous unlikely sites as the average of the counts observed for unambiguous DNA bases at those same sites. Likely sites received the remaining counts, distributed according to the relative proportions of unambiguous DNA bases. For the combined blunt and staggered models targeting primarily between nucleotides 3–4 and 4–5 upstream of the PAM (non-target DNA), we applied the above strategies for blunt and staggered models respectively to these sites. Since the ratios between blunt and staggered were not known for the remaining sites, we used the simpler estimate based on relative proportions of unambiguous DNA base insertions. Finally, we compared the estimates against expected distributions (Figs. 5a, b).

**Re-analysis of Cas9-induced DNA end structure data**. We downloaded targeted sequencing data of DNA ends induced by Cas9 targeting using the SRA-toolkit, after selection using the NCBI SRA Run Selector, specifically runs SRR1617071 to SRR1617082 in project PRJNA264361. The assays involved Cas9 cleavage guided to a region on chromosome 6 in mouse Ligase IV-deficient pre-B cells arrested in G1 phase[33]. Upon Cas9 targeting, top and bottom DNA ends were ligated into a hairpin and subject to bisulfite treatment to facilitate amplification for high-throughput DNA sequencing. We aligned the sequences in each file against the wild-type hairpin sequence assuming blunt ends between nucleotides 3–4 upstream of the PAM. For this purpose, we used BLASTn with the following options -max_target_seqs 500000 -max_hsps 1 -num_threads 30 -outfmt "6 qacc sacc qstart qend sstart send evalue bitscore length pident nident mismatch positive gaps ppos btop". We filtered low quality sequences and primers by keeping alignments at least 50-bp long containing at most 5 gaps. We also filtered sequences whose alignment did not span a region of at least 50-bp around the expected target site. We then used both the starting position of each alignment in the reference sequence and the BLASTn trace-back operations (BTOP) to isolate the pattern of changes relative to wild-type within a region of 30-bp around the expected target site. In parallel, we generated mutation patterns for all possible DNA end conformations within a target region of 30-bp around the expected target site. Finally, we matched the generated mutation patterns against the patterns extracted from the HCoDES data to identify the corresponding DNA end structures. Note that many of the HCoDES sequences contained substitutions at random positions, primarily TC, possibly also due to failed bisulfite conversion. Since substitutions did not show consistent patterns associated with particular DNA end conformations and had no impact in the ranking of top observed patterns, we decided to ignore them so that they would not confound otherwise valid matching patterns.

**Reporting Summary**. Further information on experimental design is available in the Nature Research Reporting Summary linked to this article.

## Data availability

High-throughput sequence data and processed data that support the findings of this study have been deposited at the NCBI GEO with accession GSE127752. The mapping and expression reads for the cell line containing 36 TRIP IRs correspond to samples GSM1207441, GSM1207452, GSM1207463, GSM1207474 in GEO series GSE48606. These are directly linked in our record GSE127752. The HCoDES sequence data were retrieved from the NCBI SRA repository as samples SRR1617071 to SRR1617082 from SRA study SRP049086. Source data for the figures in this manuscript and in the Supplementary Information file are available in the Source Data file. All other relevant data are available from the authors upon reasonable request.

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

## Acknowledgements

We thank the NKI Genomics Core Facility and the NKI Flow Cytometry Facility for their technical support. We also thank Anton Berns, Heinz Jacobs, Hein te Riele, Tim Harmsen, Eva Brinkman and Bas van Steensel for helpful discussions and/or critical reading of the manuscript. This research was supported by a Netherlands Organization for Scientific Research (NWO) grant 823.02.007 to W.A. and M.v.L. A.V.P. was supported by an European Research Council (ERC) Advanced Grant 293662 to Bas van Steensel.

## Author contributions

S.G., J.P.G., W.A., A.V.P., L.F.A.W. and M.v.L. designed the research. S.G., W.A. and A.V.P. performed the experimental work. W.A. and J.d.J. preprocessed the TRIP and epigenomic data. J.P.G. designed and performed the subsequent data analysis, and made most figures. S.G., J.P.G. and W.A. interpreted the results. S.G. and J.P.G. wrote the manuscript. L.F.A.W. and M.v.L. revised the manuscript, and supervised the analytical and experimental research.

## Additional information

**Competing interests:** The authors declare no competing interests.

