## [Peer Review File · Nature Communications]

Reviewers' Comments:

Reviewer #1:

Remarks to the Author:

General comments

1. There are interesting aspects in the work by Gisler et al, most notably, the conclusion that recombinant target sequences (IR, integrated reporter) scattered throughout the genome of murine ESCs, can engage with and be cleaved by Cas9:gRNA complexes to different extents. Less surprisingly, the gRNA sequences also had an impact on the mutation frequencies and indel profiles at the exogenous (i.e. EGFP) sequences targeted.

2. The methods and analysis employed by the authors are generally robust.

3. It is noteworthy pointing out, however, that their findings were exclusively obtained through experiments carried out in rather surrogate cellular models based on the ectopic integration of ~1-kb EGFP reporter cassettes into endogenous genomic sequences. In addition, the findings were derived from cells subjected to ~25-36 Cas9-induced DSBs. These findings are difficult to extrapolate to a "physiological" gene editing context in which, for the most part, only a pair of targeted DSBs are normally made.

Although it would not have solved the latter "supra-physiological" DSB formation issue, I wonder whether the authors considered the alternative/complementary approach of targeting endogenous same-sequence sites located in judiciously chosen polymorphic (hence tagged) repetitive sequences/elements or multiple-copy genes.

Major comments

1. In the Abstract, after writing : "Understanding the impact of guide RNA (gRNA) and genomic locus on CRISPR/Cas9 activity is crucial to design effective gene editing assays.", the authors state: " However, it is challenging to profile Cas9 cleavage in the endogenous cellular environment. Here, we leveraged our TRIP technology to integrate ~1k barcoded reporter genes in the genomes of mouse embryonic stem cells" And, subsequently they go on writing: "We found that genomic locus and gRNA-sequence explained most variation in mutation efficiency." These sentences bring to the front a contradiction-in-terms in that, to assess gene editing events "in the endogenous cellular environment" the authors chose to target exogenous expression units whose effects on the endogenous cellular environment are unknown/unpredictable. Indeed, the "genomic locus" they refer to elsewhere in the text and in the sentence: "We found that genomic locus and gRNA-sequence explained most variation in mutation efficiency.", actually refers to an exogenous-endogenous composite/hybrid locus that does not necessary reflect the transcriptional, genomic and epigenomic (TGE) features of the pre-existing native locus (see 2.). This should be better clarified and the aforementioned potential caveats mentioned.

2. The authors conclude that: "The variation in mutation frequency across loci correlated weakly with TGE (Transcription, Genomic, Epigenomic) features, in contrast with previous studies in vitro showing significant effects of nucleosome occupancy and chromatin organization on Cas9 binding and cleavage efficiency^{18, 19}". However, they cannot exclude the possibility that the integration of an entire foreign expression unit (IR, integrated reporter) results in the disruption of TGE features linked to the native endogenous locus, effectively erasing otherwise contributing effects of pre-existing TGE features (e.g. transgenic enhance/promoter elements have been shown to alter endogenous gene regulation via local and/or long range effects). In other words, it is far from given that the endogenous context dominantly impinges its transcriptional and epigenetic features upon the exogenous DNA whose sequences are targeted by the author's 3 Cas9:gRNA complexes. The other way around might in fact be the prevalent scenario in a genomic site-dependent manner.

Thus, at least some key TGE features (indicated in Figure 3) should be experimentally checked at independent, randomly-selected, IR sites in model cells and parental mES cells for establishing whether or not IR+ regions maintain the TGE features characteristic of the respective, pre-existent, native loci.

3. Related with the previous point. The PGK promoter as well as the other recombinant enhancer/promoters present in each IR are likely to cross-talk in an integration site-dependent manner with endogenous regulatory sequences and other cis-acting elements. Integration site-dependent mechanisms one can envision are: DNA looping, acquisition of different epigenetic marks, sense or anti-sense transcription in relation to that of endogenous genes, heterologous polyadenylation etc. all making it difficult the filtering and interpretation of the data. Albeit arguably not full-proof, possible ways of "normalizing" for these differences, would be to run experiments in independently isolated clones varying in their IR integration profiles and including a promoter-less (and polyadenylation signal-less) reporter ORF.

4. From the above it follows that the opening sentence in the Conclusions section: "Here, for the first time, we characterized Cas9-induced mutation signatures at thousands of endogenous loci across the regulatory landscape of mES cells." Is formally incorrect. By the same token, the claim at the end of the Conclusions section that reads: "We demonstrated that CRISPR-on-TRIP is a robust tool to profile Cas9 activity in the endogenous cellular environment, and can be combined with other assays to study the influence of a variety of processes on Cas9 activity (Figure 7)." Is formally incorrect.

5. The authors do not consider the possibility that a fraction of their sequence mutation reads can be the result not of simple end-to-end DSB ligation but of long-range translocations likely to be triggered by the >25 DSBs generated on a per cell basis. Such events will skew their data, e.g., by associating IR barcodes with different promoters or "mis-call" TGE features. Were these translocation events filtered (e.g. computationally by identification of accurately ligated distal sequences)? If not, an idea about the frequencies of translocations generated in their mCherry+/Cas9:gRNA-transfected cells (e.g. FISH analysis, translocation junction sequence analysis or WGS) should, in this regard, be very informative.

6. The authors make an extensive series of conjectures under the sub-section of the Results titled: "Combined blunt and staggered model explains insertion counts" regarding the processes yielding the observed EGFP indel patterns. In their reasoning, they do not include the possibility that, in addition to: "Besides direct re-ligation, blunt ends may be processed leading to a deletion or, crucially, template-independent addition of nucleotides", there might equally exist a biased selection for specific indels which, by disrupting cyclic Cas9:gRNA re-cleavage of accurately ligated DNA ends, become "fixed" in the population. This is a likely possibility in view of the fairly long-term nature of the experiments involving substantial cell division, sorting and expression of Cas9:gRNA complexes from long-lived DNA templates.

7. The additive informative value of the results described under sub-section: "Cas9 induces blunt and staggered DNA ends with 1-nt 5'overhangs in mouse pre-B cells" is moderate at best, also in view of the rather specific and artificial cellular system i.e. pre-B cells that are (i) Ligase IV-deficient and (ii) arrested in G1 (with which drug ? can pleiotropic effects of the drug be excluded ?)

8. Neither in Figure 1 nor elsewhere is there mentioning or indication of which polyadenylation signal(s) is associated with the reporter expression units. Could these terminator elements be an additional contributing factor to a change in the TGE features of a genomic locus after reporter cassette integration?

Minor comments

1. The first section of the Results lacks a sub-heading. Moreover it reads as a generic introductory paragraph amassing the description of different experiments without referring to Figures/Results. It might be somewhat trimmed and/or split and incorporated in text pointing to actual data.
2. Figure 1C, right panel. The legend states: "In TRIP pool assays, Cas9 was complexed with sgRNA2 or sgRNA3 (right). Knock-in of a single-stranded oligodeoxynucleotide (ssODN) was performed with sgRNA2." However, the corresponding diagram gives the impression that the ssODN was tested with all 3 guides. Perhaps better, relocate the ssODN seq. next to sgRNA2 as opposed to on top of the stack.
3. To facilitate understanding of the methodology and interpretation of the results, briefly, described the methods indicated in: "Cell culture and TRIP library construction and transfection were performed according to the TRIP protocol^{12, 13}."
4. Idem for: "Barcode abundance was also estimated by sequencing genomic DNA (gDNA), and used to normalize IR expression. These procedures were as described in the TRIP protocol^{12, 13}."
5. No experimental details on: "To avoid reduced transfection efficiency of mCherry-expressing Cas9-sgRNA plasmid and ssODN with Lipofectamin 2000, we followed the protocol from Nucleofector™ Kit for Mouse Embryonic Stem Cells (Lonza) instead."
6. Next to their multiplexed DSB-induced indel frequencies scored by sequencing methods, the authors could, I believe, easily add as an independent quality control check for differential gRNA activity, plot the reduction in EGFP MFI in their mCherry-sorted populations.

Reviewer #2:

Remarks to the Author:

Summary

The manuscript entitled "Multiplexed Cas9 targeting reveals genomic location effects and gRNA-based staggered breaks influencing mutation efficiency" by Gisler et al. employed a TRIP strategy to dissect variables affecting SpCas9 editing in mouse embryonic stem cells. By targeting identical target sequences in parallel in 36 different genomic regions in an engineered clonal cell line and in about 25 locations in cell pools, the authors maximized the power to quantify genomic location effects on SpCas9 editing efficiency. The results are convincing and consistent with others' observations or assumptions on genomic location effect using different methods. The high number of insertions in these cells may have altered local chromatin structures and rendered the regulatory element association analysis less sensitive. The authors further dissected the mutational profiles derived from three different sgRNAs (sgRNA 1 and sgRNA 2 guide sequences are almost completely complementary, however) and elegantly analyzed the wobbling cleavage nature of SpCas9 that generates either blunt ended cleavage or staggered cleavage with 1-nt 5' overhangs. All data analysis was thorough and sound and each paragraph by itself was well-written. Overall, this is a high quality study, but there is short of novelty and significance for publication in Nature Communications, notwithstanding that the use of the TRIP strategy is unique.

Comments

1. All variables were measured from mCherry-sorted cells to increase the overall efficiency and the data represented population averaging effects. This is a common practice. However, it would make the study more interesting to also analyze individual cells (after some expansion) or sub populations of cells sorted by different mCherry intensities (representing different Cas9 RNP levels) or by different GFP intensities (representing different editing levels). This could provide insights into how Cas9 RNP concentration affects efficiency, mutation profile, and cellular double stranded break repair choice. The authors also can make use of the TRIP strategy to test whether target

copy number is a factor affecting the efficiency on individual targets and cellular double stranded break repair choice.

2. The authors provided a detailed analysis on the editing efficiency and mutation pattern from three sgRNAs. sgRNA 3 was more efficient and far more frequent in generating staggered cleavage than either sgRNA 1 or sgRNA 2. This is an interesting observation. However, the number of sgRNAs tested is very limited and thus it is not certain what the differentiating factor is: the %GC, the sequence context, or the local DNA helical structure? Testing more sgRNAs with different sequence features may help to pin down the major determinant, which would have practical implication, such as helping one to design sgRNA to favor the 1-nt insertion mutation.

3. The deletion mutation patterns in Figure 4B seem to suggest non symmetrical deletion distributions from the expected cleavage positions, biased toward the PAM distal side. Plotting base positional mutation frequency from all data points could make the patterns more conspicuous and conclusive as to whether deletion indeed more likely to occur on the PAM distal side after SpCas9-induced double stranded break.

4. Each paragraph of the manuscript was well written by itself, but the manuscript as a whole seems somewhat fragmented. It could be further improved structurally to emphasize more important results and help the reader to make a better connection.

Response to Reviewers on Manuscript NCOMMS-18-13357-T.

Reviewer #1 (Remarks to the Author):

General comments

1. There are interesting aspects in the work by Gisler et al, most notably, the conclusion that recombinant target sequences (IR, integrated reporter) scattered throughout the genome of murine ESCs, can engage with and be cleaved by Cas9:gRNA complexes to different extents. Less surprisingly, the gRNA sequences also had an impact on the mutation frequencies and indel profiles at the exogenous (i.e. EGFP) sequences targeted.

As the reviewer mentions, one of the main conclusions of our study is that IRs integrated throughout the genomes of mES cells can be targeted and cleaved by Cas9:gRNA complexes, importantly resulting in locus-specific mutation efficiencies (Figures 2D, 2G). We show that some loci are more susceptible to mutate than others upon Cas9 cleavage, even though we are not able to pin down the major factors influencing such susceptibility at this point.

Another important conclusion is not necessarily that gRNA sequence influences mutation frequency and indel profile, but rather the result that gRNA sequence determines both insertion efficiency (Figures 2A, 2B, 2G) and pattern (Figure 4) specifically through a fraction of DSBs with staggered DNA ends (Figure 5). We present evidence that the observed insertion patterns result from a specific staggered conformation (1-nt 5' overhang between nucleotides 314 and 415 upstream of the PAM). We show that the insertion patterns are consistent with the occurrence of such staggered DNA ends, in addition to blunt DNA ends (Figure 5B). Moreover, we show that the 1-nt 5' conformation can be found in independent Cas9 targeting in LigaseIV-deficient mouse pre-B cells in vitro (Figure 5C).

This result could be influential for Cas9-based gene editing. Staggered ends are preferred for gene editing, since they allow control over the orientation of an insert into the host DNA. Knowledge of the exact DSB conformation can be used to design specific DNA inserts, and ultimately improve Cas9-based editing accuracy and efficiency. Although more extensive studies with larger numbers of gRNAs will be needed in order to assess the contribution of specific gRNA sequence properties, as mentioned by Reviewer #2, our study constitutes a step forward to improve control over Cas9-based gene editing techniques.

2. The methods and analysis employed by the authors are generally robust.

3. It is noteworthy pointing out, however, that their findings were exclusively obtained through experiments carried out in rather surrogate cellular models based on the ectopic integration of 1-kb EGFP reporter cassettes into endogenous genomic sequences. In addition, the findings were derived from cells subjected to 25-36 Cas9- induced DSBs. These findings are difficult to extrapolate to a “physiological” gene editing context in which, for the most part, only a pair of targeted DSBs are normally made.

Although it would not have solved the latter “supra-physiological” DSB formation issue, I wonder whether the authors considered the alternative/complementary approach of targeting endogenous same-sequence sites located in judiciously chosen polymorphic (hence tagged) repetitive sequences/elements or multiple-copy genes.

Regarding the concern that our findings are based on cellular models with 25-36 Cas9-induced DSBs per cell, we note that most results in our manuscript are consistent with data reported in other studies comprising only a few DSBs per cell (Figures 2, 4) (Brinkman et al., NAR 2014; Liao et al., NAR 2015; van Overbeek et al., Molecular Cell 2016). Our main conclusions are further supported by analyses of those frequencies and patterns, as well as analyses of independent data (Figures 5, 6), which both reviewers accept or approve.

The one observation we believe stood out is that we do not see a clear correlation between transcriptional, genomic and epigenomic (TGE) features and mutation frequency (Figures 3B, 3C, 3D). We believe that we may not have been clear and note that the literature is actually scarce in this regard, since most previous work has focused on the impact of (epi)genomic context on Cas9 binding, rather than mutation efficiency. It is therefore unclear whether we should expect a good correlation between TGE context and Cas9-induced mutation efficiency. Specifically, we found one study reporting an effect on mutation frequency at engineered targets upon exogenous manipulation of chromatin accessibility using doxycycline (Chen et al., NAR 2016). Another study on hundreds of off-target sites of four guide RNAs, however, found that native chromatin accessibility and DNA methylation were predictive of Cas9 binding, but mutation frequency did not correlate well with Cas9 binding at the surveyed sites (Wu et al., Nature Biotechnology 2014). Our findings are consistent with the latter study. It is a possibility that the TGE landscape affects Cas9 binding and cleavage, but that this is masked down the line by stochasticity in the outcome of DNA repair. While we do not have a final explanation, we have carefully considered the reviewer’s concerns below and concluded that our data does not support the suggestion of significant perturbations to the TGE landscape upon reporter cassette integration or frequent translocations following Cas9 targeting (please see answers to the major comments below, namely 2, 3, and 5). In the new version of the manuscript, we have extended the sentences referring to the above publications in the Introduction to clarify their contribution and the knowledge gap (see highlighted text in the last but one paragraph of the Introduction). We have also extended the Discussion to consider the results of both publications (see highlighted text in the Discussion). Follow-up studies will be needed to clarify this further.

In general, we agree that targeting endogenous sequences would be preferred in order to preserve the native conditions. We did consider alternatives to scale up the number of Cas9 targets, which we briefly mentioned in the Introduction of our manuscript: (i) compromising on the specificity of guide RNAs, while accepting the risk of increased off-target effects; (ii) using multiple guide RNAs, which would introduce heterogeneity among targets. None of these is desirable. The suggestion made by the reviewer of targeting repetitive sequences could be an alternative to (i) to increase the number of targets per guide RNA. However, as the reviewer mentions, it would not resolve the supra-physiological DSB formation issue. In particular, targeting thousands of loci in this way would mean cleaving at ~1k loci in every cell, which could cause serious genomic instability. Moreover, targeting repetitive sequences poses additional challenges. First, it is unclear whether results on repetitive sequences would generalize to other kinds of sequences. Choosing certain types of genomic regions like this could bias subsequent analyses. Ideally, we would like to target

a large number of regions to have sufficient statistical power, but also cover diverse TGE contexts. Second, targeted sequencing to profile Cas9 activity and mutations would require the design of at least one specific primer per “tagged” genomic region, causing scalability issues and variability in estimated mutation frequencies due to differences in primer efficiency. Alternatively, using whole genome sequencing would not resolve the challenges inherent to assembling and calling mutations in repeat-rich regions due to mapping ambiguities. In this context, the integration of reporter cassettes throughout the endogenous genomes allowed us to target a large number of loci across multiple cells, while keeping the number of targeted loci per cell relatively low and the variation due to experimental procedures under control.

Major comments

1. In the Abstract, after writing : “Understanding the impact of guide RNA (gRNA) and genomic locus on CRISPR/Cas9 activity is crucial to design effective gene editing assays.”, the authors state: “ However, it is challenging to profile Cas9 cleavage in the endogenous cellular environment. Here, we leveraged our TRIP technology to integrate ~1k barcoded reporter genes in the genomes of mouse embryonic stem cells” And, subsequently they go on writing: “We found that genomic locus and gRNA-sequence explained most variation in mutation efficiency.” These sentences bring to the front a contradiction-in-terms in that, to assess gene editing events “in the endogenous cellular environment” the authors chose to target exogenous expression units whose effects on the endogenous cellular environment are unknown/unpredictable. Indeed, the “genomic locus” they refer to elsewhere in the text and in the sentence: “We found that genomic locus and gRNA-sequence explained most variation in mutation efficiency.”, actually refers to an exogenous-endogenous composite/hybrid locus that does not necessary reflect the transcriptional, genomic and epigenomic (TGE) features of the pre-existing native locus (see 2.). This should be better clarified and the aforementioned potential caveats mentioned.

We agree with the reviewer that the use of the term “endogenous” is imprecise when referring to reporter integration loci, and have therefore clarified this throughout the text. Specifically, we replaced occurrences of “genomic locus” in the text and figures by “IR locus” (i.e. integrated reporter locus). We have also included an explicit mention to the fact that these are hybrid exogenous-endogenous loci (see Introduction).

We did cover potential caveats of exogenous-endogenous hybrid loci in the manuscript before. In particular, the issue raised by the reviewer concerning the preservation of TGE features upon reporter cassette integration, was mentioned in both the Results and the Discussion (please see response to 2. for additional details). In the new version, we have extended and improved the text to make our points clearer in this regard.

2. The authors conclude that: “The variation in mutation frequency across loci correlated weakly with TGE (Transcription, Genomic, Epigenomic) features, in contrast with previous studies in vitro showing significant effects of nucleosome occupancy and chromatin organization on Cas9 binding and cleavage efficiency^{18, 19}”. However, they cannot exclude the possibility that the integration of an entire foreign expression unit (IR, integrated reporter) results in the disruption of TGE features linked to the native endogenous locus, effectively erasing otherwise contributing effects of pre-existing TGE features (e.g. transgenic enhance/promoter elements have been shown to alter endogenous gene regulation via local and/or long range effects). In other words, it is far from given that the endogenous context dominantly impinges its transcriptional and epigenetic features upon the exogenous DNA whose sequences are targeted by the author’s 3 Cas9:gRNA complexes. The other way around might in fact be the prevalent scenario in a genomic site-dependent manner.

Thus, at least some key TGE features (indicated in Figure 3) should be experimentally checked at independent, randomly-selected, IR sites in model cells and parental mES cells for establishing whether or not IR+ regions maintain the TGE features characteristic of the respective, pre-existent, native loci.

We did not exclude the possibility that the insertion of exogenous DNA could cause changes to the TGE context at the integration loci. Specifically, the issue was mentioned at two points in the manuscript: (i) section “Regulatory landscape associates weakly with mutation frequency” of the Results; (ii) Discussion, including the excerpt transcribed by the reviewer. The first section includes the following statements:

“Mutation frequency correlated weakly with TGE features (Pearson’s $r = [-0.0779, 0.095]$, Figures 3B-3D), even though TGE features associated strongly with IR expression ($r = [-0.603, 1.0]$, Figure 3D).”

“The high correlation between IR expression and TGE features was consistent with the literature²³. This indicated that the integration of reporter genes might not have significantly altered the regulatory landscape of mES cells and suggested that the weak association of IR mutation frequency with TGE features could be due to additional factors.”

In this context, we also assessed one key TGE feature, expression, for all IRs in TRIP cells (“IR expression” or “Reporter expression” in Figure 3D). Expression is known to be well correlated with TGE features, reflects changes to the TGE landscape in general, and can therefore be used as a proxy for the remaining TGE features. The top row of the heatmap in Figure 3D shows that IR expression in TRIP mES cells correlated strongly with endogenous TGE features of wild-type mES cells. This would not be expected if the TGE landscape had been significantly perturbed upon reporter cassette integration. In any case, eventual changes to the TGE landscape would affect both IR mutation frequency and IR expression, since both were measured in TRIP cells. The fact that IR expression correlated weakly (and not better than wild-type TGE features) with IR mutation frequency (Figure 3D), despite identical TGE landscape, does not support the reasoning that changes to the TGE landscape caused by the integration of TRIP reporters could be the main source of the weak correlation. We have improved the text in the Results and Discussion sections to clarify these points (see highlighted text in subsection “Regulatory landscape associates weakly with mutation frequency”, and in the Discussion).

Further prompted by the reviewer’s comments, we included additional analyses in supplementary material. We looked into the mutation frequencies of two sets of IRs showing either high or low expression in both wild-type mES cells and in TRIP mES cells (Supplementary Figures S4, S5). The idea is that the TGE landscape could be better preserved for loci showing consistent high or low expression across wild-type and TRIP mES cells. Assuming an association between TGE features and IR mutation frequency, we would expect to see a difference in the mutation frequencies of IRs with high expression compared with IRs with low expression. However, this was not the case (Supplementary Figure S5). In summary, the selection of loci with high/low expression in both wild-type and TRIP mES cells did not result in a clear(er) association between IR mutation frequency and expression. This observation supports our previous findings and the hypothesis that additional factors could play a role in determining mutation frequency.

3. Related with the previous point. The PGK promoter as well as the other recombinant enhancer/promoters present in each IR are likely to cross-talk in an integration site-dependent manner with endogenous regulatory sequences and other cis-acting elements. Integration site-dependent mechanisms one can envision are: DNA looping, acquisition of different epigenetic marks, sense or anti-sense transcription in relation to that of endogenous genes, heterologous polyadenylation etc. all making it difficult the filtering and interpretation of the data. Albeit arguably not full-proof, possible ways of “normalizing” for these differences, would be to run experiments in independently isolated clones varying in their IR integration profiles and including a promoter- less (and polyadenylation signal-less) reporter ORF.

The inclusion of multiple promoters in our experimental design allowed us to test for promoter effects on IR mutation frequency across all IRs, that is, IR-independent promoter effects. We showed that, while such effects might exist, they appeared not to be very influential (Figures 2C and 2G).

The reviewer asks a different question, namely whether there could also be IR-dependent promoter effects. In our study, these effects fall within the category of IR locus effects (see variance explained by “locus” in Figure 2G). For completeness, we have mentioned this explicitly in the new version of the manuscript (see highlighted text in subsection “Impact of locus, guide, ssODN and promoter on mutation frequency”). We cannot distinguish IR-dependent promoter effects from other IR-dependent effects using the current experimental design. However, as in the previous point, both IR mutation frequency and IR expression were measured in TRIP mES cells and thus should both reflect any potential changes to the TGE landscape, including IR-dependent promoter effects. If eventual changes to the TGE landscape were a major source of the weak correlations between IR mutation frequency and TGE features measured in wild-type mES cells, we should have seen higher correlations between IR mutation frequency and IR expression which were both measured in TRIP cells and thus had identical TGE landscape. The fact that we did not observe this does not support the reasoning that IR-dependent changes could be a main source of observed weak correlations between IR mutation frequency and TGE features.

We also respectfully note that performing experiments in independently isolated clones with varying TRIP integrations would not be sufficient to assess or correct for potential IR-dependent promoter cross-talk effects. For this purpose, we would need to assess IRs with different promoters integrated at the same site (e.g. in different cells). As TRIP reporters integrate randomly, it is unlikely that IRs with different promoters systematically integrate at the same location across TRIP populations or clones. A potential solution could be to generate integrations with different promoters at predetermined rather than random loci. This would however require an extensive number of additional experiments without necessarily addressing the main cause of the observed correlations, and is therefore beyond the scope of this manuscript.

4. From the above it follows that the opening sentence in the Conclusions section: “Here, for the first time, we characterized Cas9-induced mutation signatures at thousands of endogenous loci across the regulatory landscape of mES cells.” Is formally incorrect.

We assume that the reviewer is referring to the hybrid endogenous-exogenous issue pointed out in major comment 1. We changed this sentence accordingly to “Here, for the first time, we characterized Cas9-induced mutation signatures at thousands of target sequences embedded across the genomes of mES cells.”

By the same token, the claim at the end of the Conclusions section that reads: “We demonstrated that CRISPR-on-TRIP is a robust tool to profile Cas9 activity in the endogenous cellular environment, and can be combined with other assays to study the influence of a variety of processes on Cas9 activity (Figure 7).” Is formally incorrect.

Again, we assume that the reviewer is referring to the hybrid endogenous-exogenous issue pointed out in 1. From points 2 and 3, we reason that the reviewer could also be doubting the robustness of the CRISPR-on-TRIP approach to profile Cas9 activity. In this context, we highlight the fact that all our measurements of mutation frequency and patterns were highly consistent and robust across experiments with varying experimental conditions. The fact that we could not detect the influence of TGE features is likely the result of biology or the experimental design, rather than the ability of the CRISPR-on-TRIP approach to profile Cas9 activity and resulting mutations. Following the reviewer’s concern, we have changed the sentence to “We showed that CRISPR-on-TRIP is a promising tool to profile Cas9 activity at a large number of sequences scattered throughout the genome, and can be combined with other assays to study the influence of a variety of processes on Cas9 activity and induced mutation patterns.”

5. The authors do not consider the possibility that a fraction of their sequence mutation reads can be the result not of simple end-to-end DSB ligation but of long-range translocations likely to be triggered by the >25 DSBs generated on a per cell basis. Such events will skew their data, e.g., by associating IR barcodes

with different promoters or “mis-call” TGE features. Were these translocation events filtered (e.g. computationally by identification of accurately ligated distal sequences)? If not, an idea about the frequencies of translocations generated in their mCherry+/Cas9:gRNA-transfected cells (e.g. FISH analysis, translocation junction sequence analysis or WGS) should, in this regard, be very informative.

We agree with the reviewer that this could be a potential concern. However, CRISPR/Cas9-induced chromosomal translocations events are rare (Jiang et al., *Sci. Rep.* 2016). Different studies seeking to optimize the frequency of translocations using RNA-guided Cas9 targeting of a pair of loci have achieved efficiencies between 1% and 2% (Lekomtsev et al., *BMC Genomics* 2016; Jiang et al., *Sci. Rep.* 2016). We note that the loci in those studies are typically involved in tumor-associated translocations and therefore could be more prone to translocate than a random pair of loci. Targeting 25-36 loci per cell could increase the frequency of translocations in general, but such events should remain rather rare for a random pair of loci. In addition, not every translocation would lead to the situation described by the reviewer. A fraction of translocation events could result in the ligation of the upstream or downstream regions of two independent DSB sites (upstream1:upstream2, or downstream1:downstream2). Such translocations would not be amplified using our protocol, and were therefore filtered automatically. Other translocation events could involve the ligation of a DSB-upstream DNA region to aDSB-downstream DNA region. We could not filter the latter type of translocations, since our sequenced amplicons covered only a region of the IR gene body (EGFP cassette), including the expected DSB site, along with the IR promoter index and the IR barcode downstream, but not the promoter itself located upstream of the IR gene (Supplementary Figure S9). The reason for this was that the complete IR sequence was 1.8-3.4kb in length, and it would be challenging to amplify reliably and consistently across ~1k IRs.

Nevertheless, we have now investigated the occurrence of translocation events for 20 randomly chosen IRs using PCR amplification and Sanger sequencing (please see Supplementary Material document). For this purpose, we sought to obtain longer amplicons spanning the IR-specific elements upstream and downstream of the expected Cas9 target site (Supplementary Figure S9). The promoter sequence was the most proximal IR-specific feature located upstream of the EGFP cassette in the IRs. As a result, we designed primers to amplify a DNA region extending from the end of the 5’LTR sequence of the IR gene (Figure S9 and Table S4: 5’LTR-amp, reporter-specific primer just upstream of the promoter) to a portion of the genomic sequence located less than 1kb downstream of the IR gene (Figure S9 and Table S4: genomic primers 1-20). Using this approach, we could check whether the locus-specific genomic region downstream of the DSB site remained associated with the correct promoter after Cas9 targeting. Any match to a different promoter than expected would indicate a translocation.

We experienced difficulties to obtain sufficient quality amplification product for sequencing, due to the combination of the following inevitable factors affecting the amplification efficiency:

- (1) long amplification products;
- (2) majority of low-abundance IRs, since each IR is only present in a subset of cells from the population;
- (3) one of the primers had to bind to genomic DNA, inherently more difficult to amplify than IR DNA;
- (4) the other primer had to bind to an LTR sequence, an otherwise abundant DNA element in mES cells, making it difficult to design specific primers; to mitigate this, we designed longer primers than usual (32nt);
- (5) varying GC content across IRs.

We were able to amplify and sequence five of the twenty IR loci (25%). Four other DNA regions were amplified, but the corresponding sequences did not match any reporter elements upstream or downstream of the expected DSB site (Table S4). This was likely caused by unspecific primer binding, since the proper amplification of the targeted IR loci using the downstream genomic primer should at least contain the sequence downstream of the DSB. Ultimately, we validated the promoter sequence for all five sequenced IRs (Table S5). Although we cannot exclude the possibility of translocations, our data (including the experiment above) do not suggest that these would be frequent nor that they would have had a major impact

on our results and conclusions thereof. We refer to these results in subsection “Regulatory landscape associates weakly with mutation frequency” of the Results.

6. The authors make an extensive series of conjectures under the sub-section of the Results titled: “Combined blunt and staggered model explains insertion counts” regarding the processes yielding the observed EGFP indel patterns. In their reasoning, they do not include the possibility that, in addition to: “Besides direct re-ligation, blunt ends may be processed leading to a deletion or, crucially, template-independent addition of nucleotides⁹.”, there might equally exist a biased selection for specific indels which, by disrupting cyclic Cas9:gRNA re-cleavage of accurately ligated DNA ends, become “fixed” in the population. This is a likely possibility in view of the fairly long-term nature of the experiments involving substantial cell division, sorting and expression of Cas9:gRNA complexes from long-lived DNA templates.

The description cited by the reviewer pertains to the outcome of a single Cas9:gRNA cleavage event, and therefore does not mention re-cleavage. We are aware that re-cleavage of accurately ligated DNA ends can occur until an indel arises and becomes nearly “fixed”, since the Cas9:gRNA complex will be much less likely to bind to and re-cleave at that site, as the reviewer suggests. This could naturally promote a biased selection for indels in general versus accurately ligated sequences, but not necessarily induce a biased selection for specific indels. During our analyses, we considered the possibility that re-cleavage could affect the ratio of blunt versus staggered ends, and thus refrained from quantifying this entirely. Our primary aim in this context was to show that both blunt and staggered cleavage occurred, regardless of their rates, and our data was suitable for this purpose. We did not explicitly mention re-cleavage events in the manuscript since they were not relevant in the context of our analysis. For completeness, we have included our above reasoning on re-cleavage in this new version of the manuscript.

7. The additive informative value of the results described under sub-section: “Cas9 induces blunt and staggered DNA ends with 1-nt 5’ overhangs in mouse pre-B cells” is moderate at best, also in view of the rather specific and artificial cellular system i.e. pre-B cells that are (i) Ligase IV-deficient and (ii) arrested in G1 (with which drug ? can pleiotropic effects of the drug be excluded ?)

In the subsections prior to the one mentioned in this point, we present evidence from analysis of mutation patterns in our sequence data that suggests the existence of staggered DSBs. Although the analytical results are compelling, it is not possible to establish whether the proposed staggered conformation really occurs prior to DSB ligation only by looking at sequence readouts. In this context, we analyze data from an independent study to investigate the claim that Cas9:gRNA targeting could produce 1-nt 5’ overhangs between nucleotides 3|4 and 4|5 upstream of the PAM. In the original study, mouse pre-B cells were purposely made Ligase IV-deficient to enable the capture of DNA ends before ligation, and therefore characterize the DNA end conformation at the break site after Cas9 cleavage. The fact that the cellular model and experimental setup differ quite significantly from the one in our study (namely in mouse cell type and cellular conditions, Cas9 complex, gRNA, experimental conditions, etc.) makes an even more compelling case as we could find back the staggered conformation proposed in our study.

In summary, the results of the subsection referred to by the reviewer show unequivocal evidence that RNA-guided Cas9 cleavage occasionally generates DSBs with 1-nt 5’ overhangs between nucleotides 3|4 and 4|5 upstream of the PAM. This can only be inferred using analytical modelling of our sequence data.

8. Neither in Figure 1 nor elsewhere is there mentioning or indication of which polyadenylation signal(s) is associated with the reporter expression units. Could these terminator elements be an additional contributing factor to a change in the TGE features of a genomic locus after reporter cassette integration?

We updated the Online Methods to include the polyadenylation signal associated with the reporter expression unit: soluble neuropilin-1 (sNRP) polyadenylation signal (see highlighted text in subsection

“Mouse embryonic stem cell culture and TRIP library transfection”). Our experimental design and generated data do not allow us to assess the potential contribution of polyadenylation signals to changes in TGE features following reporter cassette integration. However, as mentioned above and reported in the manuscript (Figure 3D), IR expression in our TRIP mES cells correlated highly with TGE features measured in wild-type mES cells. This result does not support the idea of significant and widespread changes to TGE features. Moreover, investigating the influence of polyadenylation signals on TGE features would not help clarify why the IR mutation frequency correlated weakly with IR expression (both quantified in TRIP mES cells with the same TGE landscape). For this reason, we believe that assessing the impact of polyadenylation signals on TGE features lies beyond the scope of this work.

Minor comments

1. The first section of the Results lacks a sub-heading. Moreover it reads as a generic introductory paragraph amassing the description of different experiments without referring to Figures/Results. It might be somewhat trimmed and/or split and incorporated in text pointing to actual data.

We added a sub-heading to the first section of the Results. This section is indeed an introductory paragraph outlining the different experiments. It does refer to Figure 1, and more specifically to the following: a representation of the TRIP reporter construct (Figure 1A), the general workflow from establishing the TRIP cell line and pool to performing the Cas9 targeting assays and mutation profiling by sequencing (Figure 1B), and finally the experimental design of the Cas9 targeting assays on the TRIP cell line and pool (Figure 1C). We believe that it is valuable to provide a summary of the general experimental design and assays in the beginning of the Results section, but we have now trimmed the text down to make it easier to read.

2. Figure 1C, right panel. The legend states: “In TRIP pool assays, Cas9 was complexed with sgRNA2 or sgRNA3 (right). Knock-in of a single-stranded oligodeoxynucleotide (ssODN) was performed with sgRNA2.” However, the corresponding diagram gives the impression that the ssODN was tested with all 3 guides. Perhaps better, relocate the ssODN seq. next to sgRNA2 as opposed to on top of the stack.

We relocated the ssODN sequence and placed it next to sgRNA2, as suggested by the reviewer (see Figure 1C, right panel).

3. To facilitate understanding of the methodology and interpretation of the results, briefly, described the methods indicated in: “Cell culture and TRIP library construction and transfection were performed according to the TRIP protocol^{12, 13}.”

We believe it is common practice to refer to the original work, rather than describing the protocol again. It is also not trivial to abbreviate the description of the protocol, as it might jeopardize reproducibility. Nevertheless, we are willing to include it if the Reviewer and Editor feel that this would be necessary.

4. Idem for: “Barcode abundance was also estimated by sequencing genomic DNA (gDNA), and used to normalize IR expression. These procedures were as described in the TRIP protocol^{12, 13}.”

Same as above (minor comment 3).

5. No experimental details on: “To avoid reduced transfection efficiency of mCherry-expressing Cas9-sgRNA plasmid and ssODN with Lipofectamin 2000, we followed the protocol from NucleofectorTM Kit for Mouse Embryonic Stem Cells (Lonza) instead.”

We added a more detailed description of the mentioned experimental procedure to the Online Methods (see highlighted text in subsection “Cloning and transfection of sgRNA-guided CRISPR/Cas9”).

6. Next to their multiplexed DSB-induced indel frequencies scored by sequencing methods, the authors could, I believe, easily add as an independent quality control check for differential gRNA activity, plot the reduction in EGFP MFI in their mCherry-sorted populations.

We did not assess the reduction in EGFP MFI in the mCherry-sorted populations, since we were not interested in the loss or gain of function of the EGFP reporter gene. Our goal was to characterize and quantify mutation profiles in detail regardless of their impact on the functionality of the reporter gene. In this context, sequencing offered a superior quantification of the outcome of Cas9 targeting for our purposes.

Reviewer #2 (Remarks to the Author):

Summary

The manuscript entitled “Multiplexed Cas9 targeting reveals genomic location effects and gRNA-based staggered breaks influencing mutation efficiency” by Gisler et al. employed a TRIP strategy to dissect variables affecting SpCas9 editing in mouse embryonic stem cells. By targeting identical target sequences in parallel in 36 different genomic regions in an engineered clonal cell line and in about 25 locations in cell pools, the authors maximized the power to quantify genomic location effects on SpCas9 editing efficiency. The results are convincing and consistent with others’ observations or assumptions on genomic location effect using different methods. The high number of insertions in these cells may have altered local chromatin structures and rendered the regulatory element association analysis less sensitive. The authors further dissected the mutational profiles derived from three different sgRNAs (sgRNA 1 and sgRNA 2 guide sequences are almost completely complementary, however) and elegantly analyzed the wobbling cleavage nature of SpCas9 that generates either blunt ended cleavage or staggered cleavage with 1-nt 5’ overhangs. All data analysis was thorough and sound and each paragraph by itself was well-written. Overall, this is a high quality study, but there is short of novelty and significance for publication in Nature Communications, notwithstanding that the use of the TRIP strategy is unique.

To the best of our knowledge, this is the first study to characterize Cas9-induced mutation frequencies and patterns at such a large number of loci (~1k) per guide RNA, and thus to assay effects of genomic location at scale. Previous studies of mutation frequencies or patterns were based on a limited number of target sites, and therefore could not probe genomic location effects. Other studies specifically looking into genomic or epigenomic context have focused mostly on the effects on Cas9 binding, rather than mutation frequencies and patterns. It therefore remains unclear whether (epi)genomic context is predictive of Cas9-induced mutation efficiency. In this regard, we note that understanding mutation efficiency is crucial to practical applications of the CRISPR/Cas9 system while binding is not necessarily so. As the reviewer points out, our study is also unique in employing a strategy like TRIP to multiplex Cas9 targeting using highly specific sgRNAs, and enable target sequence-independent analysis of mutation frequencies and patterns.

In our study, we show that IR locus is a key determinant of mutation frequency, but IR mutation frequency correlates weakly with regulatory elements at such locus. Regarding the concern expressed by the reviewer that the integration of TRIP reporters may have rendered the latter analysis less sensitive, we believe that we were not sufficiently clear in our manuscript in pointing out that IR expression was measured in TRIP cells. While we accept that the integration of TRIP reporters may have induced changes to regulatory elements (TGE landscape or features in our manuscript), we note that both IR mutation frequency and IR expression were quantified in cells with the same TRIP integrations (thus identical TGE landscape, possibly reflecting eventual changes), and did not show a stronger association than IR mutation frequency and TGE features of wild-type mES cells (Figure 3D). This observation does not support the idea that eventual

changes to the TGE landscape were the main source of the weak association between IR mutation frequency and TGE features. For additional details, we refer to our response to major comment 2 by Reviewer 1.

Related to the above paragraphs, we were originally surprised that we did not see a correlation between IR mutation frequency and TGE features and discussed the results from this perspective. The reason for this was that most literature reports effects of (epi)genomic context on Cas9 binding as a proxy for Cas9 activity and editing efficiency. This assumes that mutation frequency correlates with Cas9 binding. However, the degree of stochasticity in the process by which binding and cleavage lead to mutation events is not understood. Specifically concerning the effect of (epi)genomic context directly on mutation frequency, we found scarce and contradicting evidence. One study reported an effect on mutation frequency at engineered targets upon exogenous manipulation of chromatin accessibility using doxycycline (Chen et al., NAR 2016). Another study, however, found that native chromatin accessibility and DNA methylation were predictive of Cas9 binding, but mutation frequency did not correlate well with Cas9 binding at the off-target sites of four guide RNAs (Wu et al., Nature Biotechnology 2014). Our findings are consistent with the latter study, which profiled Cas9 binding, mutation frequency and epigenomic features at native genomic loci. Ultimately, it is a possibility that the TGE landscape affects Cas9 binding and cleavage, but that these effects are masked down the line by the outcome of DNA repair. We have extended the Introduction and Discussion to explain these publications in more detail, and discuss our results accordingly.

Ultimately, our work reveals effects of genomic location on mutation frequency and presents an extensive overview of associations between mutation frequency and a range of regulatory elements. Moreover, it raises important questions on the impact of the (epi)genomic context on mutation frequency that will inform and likely motivate follow-up studies.

Additionally, we show that Cas9-mediated insertion efficiency is driven by the frequency of breaks with staggered DNA ends (1-nt 5' overhang between nucleotides 314 and 415 upstream of the PAM), which is in turn determined by the guide RNA. This suggests that guide RNAs can be designed to maximize insertion efficiency in the future. Moreover, knowledge of the staggered DSB conformation can be used to design specific DNA inserts, and further improve Cas9-based editing accuracy and efficiency.

In our opinion, the knowledge put forward in this study is valuable on its own. Sharing it with the scientific community is important, as it will open the way for efforts to clarify the impact of (epi)genomic context on Cas9-based mutation efficiency, and predict insertion efficiency based on guide RNA sequence to inform guide RNA design in the future.

Comments

1. All variables were measured from mCherry-sorted cells to increase the overall efficiency and the data represented population averaging effects. This is a common practice. However, it would make the study more interesting to also analyze individual cells (after some expansion) or sub populations of cells sorted by different mCherry intensities (representing different Cas9 RNP levels) or by different GFP intensities (representing different editing levels). This could provide insights into how Cas9 RNP concentration affects efficiency, mutation profile, and cellular double stranded break repair choice. The authors also can make use of the TRIP strategy to test whether target copy number is a factor affecting the efficiency on individual targets and cellular double stranded break repair choice.

We did originally perform assays on five populations of cells sorted by different mCherry intensities (samples 1 to 5, with increasing Cas9 concentration). These assays were mainly used to fine tune the experimental protocol, and thus did not cover all the conditions of the final experiments described in the main manuscript. Specifically, we used a population of mES cells containing varying TRIP integrations

with a single promoter, PGK. Moreover, we targeted the IR loci in the cells of samples 1 to 5 using Cas9 complexed with sgRNA3 only. Following this reviewer's comment, we have included the analysis of these samples in supplementary material. In summary, we saw that Cas9 RNP concentration influenced the overall mutation efficiency, but seemed to have negligible effect on the ratio between deletion and insertion frequencies, and on mutation sizes and profiles. For completeness, we summarize the findings below.

Mutation frequency:

The overall mutation frequency varied with Cas9 concentration, as expected (Figure S7). Despite this, the ratio between deletion and insertion frequencies remained stable (Figure S7) and was consistent with the distributions observed for IRs with the PGK promoter in the multi-promoter TRIP pool (Figure 2C). Changes to these frequencies were very subtle and mostly noticeable using extreme variations in Cas9 concentration (Figure S7, sample 5 versus the rest). Both deletion and insertion frequencies increased with Cas9 concentration, with insertions seemingly more frequent than deletions at the highest Cas9 concentration (sample 5). In this regard, we note that higher Cas9 concentrations resulted in less sequenced material and lower read counts per IR, possibly due to cell death by toxicity or damage to the DNA caused by the extensive Cas9 cleavage. This could also bias the data towards less damaging mutations (e.g. cells carrying 1bp insertions could have better survival than those affected by larger deletions).

Mutation sizes and patterns:

The 5 samples at different Cas9 concentrations showed similar indel size distributions and patterns (Figure S8), and further corroborated the results in the main manuscript (Figures 4A, 4B, 4D). The most common deletion sizes were {2,4,5}-bp, and 1-bp was also by far the most frequent insertion size. The most inserted nucleotide was T, accounting for over 93% of 1-bp insertions in all 5 samples. The ten most frequent deletions were also in line with those obtained for the TRIP cell line in the main manuscript (Figure 4B). Samples 1 and 2 showed higher frequencies than the remaining three samples for CGTAT (rank 1 in Figure 4B) and TATGCG (rank 7 in Figure 4B) deletions, along with 6-bp size deletions. We note that each of these deletions represents a small fraction of the total, and therefore some variability was to be expected.

Finally, we thank the reviewer for the idea of using TRIP to assess the effect of copy number on mutation frequency, which we believe should be addressed in future work.

2. The authors provided a detailed analysis on the editing efficiency and mutation pattern from three sgRNAs. sgRNA 3 was more efficient and far more frequent in generating staggered cleavage than either sgRNA 1 or sgRNA 2. This is an interesting observation. However, the number of sgRNAs tested is very limited and thus it is not certain what the differentiating factor is: the %GC, the sequence context, or the local DNA helical structure? Testing more sgRNAs with different sequence features may help to pin down the major determinant, which would have practical implication, such as helping one to design sgRNA to favor the 1-nt insertion mutation.

We agree with the reviewer that experiments with additional sgRNAs are needed to determine which sgRNA sequence properties associate with insertion frequency. However, this will require the assessment of an extensive number of sgRNAs (in the order of hundreds or thousands). The experimental design used in this study currently limits the number of sgRNAs that could be designed against the EGFP reporter, since they would need to fit within the region covered by the sequenced amplicon. For these reasons, we believe that the suggestion made by the reviewer lies beyond the scope of this work.

3. The deletion mutation patterns in Figure 4B seem to suggest non symmetrical deletion distributions from the expected cleavage positions, biased toward the PAM distal side. Plotting base positional mutation frequency from all data points could make the patterns more conspicuous and conclusive as to whether deletion indeed more likely to occur on the PAM distal side after SpCas9-induced double stranded break.

We did notice the apparent bias of deleted regions toward the PAM distal site mentioned by the reviewer. However, as with insertions, ambiguities in the sequence make the analysis of these patterns rather challenging, and we decided not to pursue the analysis further at this time. For instance, with sgRNA1 the most common CGG deletion could comprise the loss of nucleotides 4-6 upstream of the PAM, as represented, or alternatively the loss of either nucleotides 1-3 upstream of the PAM or the PAM itself. Likewise, the most common G deletion using sgRNA2 could encompass the loss of either nucleotide 3 or 4 upstream of the PAM. It is not clear which of the positions would be more likely in each case, and we prefer not to draw conclusions in this context.

4. Each paragraph of the manuscript was well written by itself, but the manuscript as a whole seems somewhat fragmented. It could be further improved structurally to emphasize more important results and help the reader to make a better connection.

We acknowledge the point made by the reviewer. Although based on the same data, our manuscript addresses multiple research questions leading to an apparent compartmentalization. We have adjusted the text to improve the connection between the different parts of the manuscript.

References:

- Chen, X. *et al.* Probing the impact of chromatin conformation on genome editing tools. *Nucleic Acids Res* **44**, 6482-6492 (2016).
Jiang, J. *et al.* Induction of site-specific chromosomal translocations in embryonic stem cells by CRISPR/Cas9. *Sci. Rep.* **6**, 21918 (2016).
Lekomtsev, S. *et al.* Efficient generation and reversion of chromosomal translocations using CRISPR/Cas technology. *BMC Genomics* **17**, 739 (2016).
Wu, X. *et al.* Genome-wide binding of the CRISPR endonuclease Cas9 in mammalian cells. *Nat. Biotechnol.* **32**, 670 (2014).

Reviewers' Comments:

Reviewer #1:

Remarks to the Author:

In their revised version, the authors have added the statement that: "Most reports suggest that genomic context influences Cas9 binding, but effects on editing efficiency have been underexamined."

And in the rebuttal letter:

"... most previous work has focused on the impact of (epi)genomic context on Cas9 binding, rather than mutation efficiency. It is therefore unclear whether we should expect a good correlation between TGE context and Cas9-induced mutation efficiency. Specifically, we found one study reporting an effect on mutation frequency at engineered targets upon exogenous manipulation of chromatin accessibility using doxycycline (Chen et al., NAR 2016)."

In contrast to what the authors write, there are more than the cited study (Chen et al., NAR 2016) showing, in cells, a clear influence of epigenomic features not only on Cas9 binding but also, importantly, Cas9 activity. These studies are:

- Knight et al. Dynamics of CRISPR-Cas9 genome interrogation in living cells. *Science*. 2015 Nov 13; 350(6262):823-6. (Study on Cas9 binding).
- Daer et al. The Impact of Chromatin Dynamics on Cas9-Mediated Genome Editing in Human Cells. *ACS Synth Biol*. 2017 Mar 17; 6(3):428-438. (Study on Cas9 activity).
- Jensen et al. Chromatin accessibility and guide sequence secondary structure affect CRISPR-Cas9 gene editing efficiency. *FEBS Lett*. 2017 Jul; 591(13):1892-1901. (Study on Cas9 activity)

More recently:

- Uusi-Mäkelä et al. Chromatin accessibility is associated with CRISPR-Cas9 efficiency in the zebrafish (*Danio rerio*). *PLoS One*. 2018 Apr 23; 13(4):e0196238. (Study on Cas9 activity).
- Yarrington et al. Nucleosomes inhibit target cleavage by CRISPR-Cas9 in vivo. *Proc Natl Acad Sci U S A*. 2018 Sep 18; 115(38):9351-9358. (Study on Cas9 activity).

Thus, the results from these studies, together with other in vitro work, are at variance with the author's conclusion that there is a weak correlation between epigenomic features at target sequences and Cas9 activity.

Conjectures and assumptions abound in the addressing of the reviewers comments in the rebuttal document. In addition, other than IR expression inferences, the authors have refrained from providing any additional supportive experimental data that Cas9 activity correlates weakly with TGE features.

The authors argue that EGFP expression levels (high and low) from randomly inserted reporter (IR) templates can be used as a "proxy" for TGE features (transcriptional, genomic and epigenomic) as a whole. However, regardless of the endogenous flanking regions, Cas9 EGFP target sequences within IRs expressing at high and low levels are expected to be both in a similarly accessible epigenetic context leading to the observed similar Cas9 activity. From this analysis the authors claim a weak correlation between TGE features and Cas9 mutagenicity. Clearly, transcriptional activity is not a proper "proxy" for epigenomic features, an important portion of which, is associated with silenced states. More relevant to cover the "E" of Epigenomic in the TGE acronym would have been to experimentally assess actual epigenetic marks spanning the EGFP Cas9 target sequences in IRs without biasing this analysis exclusively to EGFP-expressing IRs.

Minor comments:

It is somewhat odd the author's reply to the earlier minor comment 3 (and 4) to briefly describe

key experimental steps.

Minor comment 3. To facilitate understanding of the methodology and interpretation of the results, briefly, described the methods indicated in: "Cell culture and TRIP library construction and transfection were performed according to the TRIP protocol^{12, 13}."

Author's answer: "We believe it is common practice to refer to the original work, rather than describing the protocol again. It is also not trivial to abbreviate the description of the protocol, as it might jeopardize reproducibility. Nevertheless, we are willing to include it if the Reviewer and Editor feel that this would be necessary."

Concluding, I reiterate my earlier general comment that the: "...findings were exclusively obtained through experiments carried out in rather surrogate cellular models based on the ectopic integration of ~1-kb EGFP reporter cassettes into endogenous genomic sequences. In addition, the findings were derived from cells subjected to ~25-36 Cas9-induced DSBs. These findings are difficult to extrapolate to a "physiological" gene editing context in which, for the most part, only a pair of targeted DSBs are normally made."

In their rebuttal the authors state: "The fact that we could not detect the influence of TGE (transcriptional, genomic and epigenomic) features is likely the result of biology or the experimental design, rather than the ability of the CRISPR-on-TRIP approach to profile Cas9 activity and resulting mutations". On the basis of the available data, I think that the cause for the authors not detecting the influence of TGE features on Cas9 activity is likely the result of the surrogate nature of the experimental system rather than biology.

Thus, I am not persuaded about the author's conclusion that epigenomic features have a weak or no influence of Cas9 mutation efficiency.

Reviewer #2:

None

Response to Reviewers on Manuscript NCOMMS-18-13357A.

Reviewer #1 (Remarks to the Author):

In their revised version, the authors have added the statement that: “Most reports suggest that genomic context influences Cas9 binding, but effects on mutation efficiency have been underexamined.”

And in the rebuttal letter: “... most previous work has focused on the impact of (epi)genomic context on Cas9 binding, rather than mutation efficiency. It is therefore unclear whether we should expect a good correlation between TGE context and Cas9-induced mutation efficiency. Specifically, we found one study reporting an effect on mutation frequency at engineered targets upon exogenous manipulation of chromatin accessibility using doxycycline (Chen et al., NAR 2016).”

In contrast to what the authors write, there are more than the cited study (Chen et al., NAR 2016) showing, in cells, a clear influence of epigenomic features not only on Cas9 binding but also, importantly, Cas9 activity. These studies are:

- Knight et al. Dynamics of CRISPR-Cas9 genome interrogation in living cells. *Science*. 2015 Nov 13;350(6262):823-6. (Study on Cas9 binding).
- Daer et al. The Impact of Chromatin Dynamics on Cas9-Mediated Genome Editing in Human Cells. *ACS Synth Biol*. 2017 Mar 17;6(3):428-438. (Study on Cas9 activity).
- Jensen et al. Chromatin accessibility and guide sequence secondary structure affect CRISPR-Cas9 gene mutation efficiency. *FEBS Lett*. 2017 Jul;591(13):1892-1901. (Study on Cas9 activity)

More recently:

- Uusi-Mäkelä et al. Chromatin accessibility is associated with CRISPR-Cas9 efficiency in the zebrafish (*Danio rerio*). *PLoS One*. 2018 Apr 23;13(4):e0196238. (Study on Cas9 activity).
- Yarrington et al. Nucleosomes inhibit target cleavage by CRISPR-Cas9 in vivo. *Proc Natl Acad Sci U S A*. 2018 Sep 18;115(38):9351-9358. (Study on Cas9 activity).

Thus, the results from these studies, together with other in vitro work, are at variance with the author's conclusion that there is a weak correlation between epigenomic features at target sequences and Cas9 activity.

We thank the Reviewer for bringing these studies to our attention, which we carefully reviewed. Three of the studies assess effects of the epigenome on Cas9 mutation efficiency (Daer et al. ACS Synth Biol 2017; Jensen et al., FEBS Lett 2017; Uusi-Mäkelä, PLoS ONE 2018), which is also the subject of our study. The other two studies address effects of the epigenome on Cas9 binding (Knight et al., Science 2015) or cleavage (Yarrington et al., PNAS 2018). Below we discuss the two sets of studies in connection with our work.

Studies on Cas9 mutation efficiency (Daer et al. 2017, Jensen et al. 2017, Uusi-Mäkelä et al. 2018; Here we provide a summary and include a more elaborate discussion of their findings at the end of this document). These studies report weak associations between epigenomic context and Cas9 mutation efficiency, in agreement with our findings. The reported effects are modest even when exogenously controlling chromatin states (Daer et al. 2017), and are dependent on the guide RNA and possibly other factors (Daer et al. 2017, Jensen et al. 2017, Uusi-Mäkelä 2018). This evidence is also consistent with the work we cite by Wu et al. (Nature Biotechnol 2014), which found that DNA accessibility was predictive of Cas9 binding, but binding correlated weakly with mutation efficiency. We added the three references to the Introduction and Discussion of our manuscript.

We note that two of the three studies provide anecdotal evidence on a limited number of sites (Daer et al. 2017; Jensen et al. 2017). In addition, Jensen et al. (2017) do not adequately isolate epigenetic from gRNA sequence effects, and the effects can therefore not be directly attributed to epigenetics. In Uusi-Mäkelä et al. (2018), public mutation efficiency and epigenome data were used to calculate correlations for a number of epigenetic features and developmental stages without correction for multiple testing. This led to an optimistic number of significant p-values, but even without correction, a small number of correlations were considered significant, in line with our data. Also in Uusi-Mäkelä et al. (2018), efficiencies of multiple gRNAs were pooled, and thus any potential differences across gRNAs such as those detected by us and Daer et al. (2017) are not apparent.

What further distinguishes our work from these studies, is the scale of the assays (~1k IRs) and use of TRIP to enable sequence-independent analysis across sites. In contrast to the cited studies, the ~1k sites across the genome result in a large sample size, enabling properly powered statistical analyses.

Studies on Cas9 binding or cleavage (Knight et al. 2015, Yarrington et al. 2018)

These studies cannot be directly compared with our work, since binding/cleavage and editing may have different dependencies on epigenomic context. Others have explicitly shown that effects on Cas9 binding may not directly translate to effects on Cas9 mutation efficiency (Daer et al. 2017, Wu et al. 2014).

Specifically, effects on binding/cleavage result from direct perturbation of the Cas9 nuclease, while effects on mutation efficiency depend on binding, cleavage and other factors that determine the outcome of (multiple) cleavage and repair events. Several factors can induce stochasticity and variation, namely: (i) distinct mutation rates inherent to repair mechanisms, (ii) stochastic outcome of DSB repair, increased by the possibility for re-cleavage and repair by different repair mechanisms; (iii) reprogramming of the epigenome during the S-phase of the cell cycle, leading to DNA being temporarily accessible in otherwise inaccessible regions. Additional factors can play a role, as suggested in the studies cited by the Reviewer.

We have added references to (Knight et al. 2015) and (Yarrington et al. 2018) to the Introduction and Discussion of our manuscript as additional literature on epigenome effects on Cas9 binding and cleavage.

They authors argue that EGFP expression levels (high and low) from randomly inserted reporter (IR) templates can be used as a “proxy” for TGE features (transcriptional, genomic and epigenomic) as a whole. We understand the concern that we did not experimentally validate epigenomic features and have thus removed the suggestion that results on expression could be extrapolated to other TGE features from the

manuscript. We further clarify that we have not used expression as a replacement for TGE features in our analysis, results, or figures.

However, regardless of the endogenous flanking regions, Cas9 EGFP target sequences within IRs expressing at high and low levels are expected to be both in a similarly accessible epigenetic context leading to the observed similar Cas9 activity. From this analyses the authors claim a weak correlation between TGE features and Cas9 mutagenicity.

The strong correlation between IR expression and TGE features of endogenous flanking regions in wild-type mES cells indicates that the expression level at the Cas9 target sites reflects the epigenomic context of the proximal endogenous flanking region located only ~100bp away. Based on the strong correlation between IR expression and TGE features, together with evidence from Wu et al (2014), we had suggested that the weak correlations between TGE features and Cas9 mutation efficiency were likely not an artifact. Since we did not validate this hypothesis, as mentioned above we removed it from the manuscript.

Regarding the statement, ‘...IRs expressing at high and low levels are expected to be both in a similarly accessible epigenetic context leading to the observed similar Cas9 activity’, we clarify that this cannot be concluded from the data. First, we cannot exclude the existence of an effect between IR expression and Cas9-induced mutation efficiency based on a statistical test delivering a non-significant result (Figure S5). Second, expression is generally known to reflect epigenetic context. Our expectation would therefore be that IRs expressing at high and low levels would correlate with the underlying epigenetic context. Granted, this was not validated but we consider this a reasonable expectation. Third, as an alternative to the suggestion made by the Reviewer, it is plausible that the association between mutation efficiency and epigenomic features is in fact weak. The studies by Wu et al. (2014), Daer et al. (2017) and Uusi-Mäkelä (2017) support this possibility by reporting weak correlations between epigenomic context and Cas9-induced mutation efficiency at endogenous sites. In particular, the studies by Wu et al. (2014) and Daer et al. (2017) show that the clear effect of epigenomic context on Cas9 binding does not necessarily translate to mutation efficiency. Daer et al. (2017) also report detectable or undetectable effects depending on guide RNA in regions subject to identical variations in the assayed epigenomic mark.

In summary, evidence in these studies points to weak effects of the epigenome on mutation efficiency, and does not support the expectation that similar Cas9 mutation efficiency implies similar epigenomic context. This relationship and the extent to which these and other factors may affect mutation efficiency remains to be clarified and needs to be further investigated.

Clearly, transcriptional activity is not a proper “proxy” for epigenomic features, an important portion of which, is associated with silenced states. More relevant to cover the “E” of Epigenomic in the TGE acronym would have been to experimentally assess actual epigenetic marks spanning the EGFP Cas9 target sequences in IRs without biasing this analysis exclusively to EGFP-expressing IRs.

As mentioned above, we agree that transcriptional activity is not a replacement for experimentally assessing epigenomic features. We understand the Reviewer’s concern about extrapolating our findings on expression to other TGE features without further validation. As previously stated, we have removed this suggestion from the main manuscript and the supplementary material.

It is not clear to us why our analysis could be biased to EGFP-expressing IRs. We have not selected IRs based on EGFP fluorescence or protein expression at any point in this study. We have selected cells based on mCherry fluorescence and analyzed all IRs. Furthermore, IR expression was not based on EGFP fluorescence or protein expression. Instead, IR expression was quantified directly for every IR as barcode counts in cDNA (RNA-seq) normalized by barcode counts in genomic DNA (DNA-seq), both of which determined using high-throughput sequencing for a region of 168bp spanning the Cas9 target sites.

Concluding remark:

On the basis of the available data, I think that the cause for the authors not detecting the influence of TGE features on Cas9 activity is likely the result of the surrogate nature of the experimental system rather than biology. Thus, I am not persuaded about the author's conclusion that epigenomic features have a weak or no influence of Cas9 mutation efficiency.

We understand the concern that results on TGE features other than expression were not experimentally validated, and changed the text accordingly. However, we did validate IR expression and our results on epigenomic features are in line with the findings of other studies (Wu et al. 2014), including those cited by the Reviewer (Daer et al. 2017; Jensen et al. 2017; Uusi-Mäkelä et al. 2018). Unfortunately it would be extremely challenging to validate epigenomic features and achieve conclusive evidence on the cause of the weak correlations between TGE features and Cas9 mutation efficiency (see below). We believe that the results on epigenomic context can be considered exploratory, since they are certainly not the main point of this manuscript. We have toned down the text accordingly. We sincerely hope that Reviewer 1 is satisfied with the changes we have made to the text to address the important points made by the Reviewer.

Reviewer 1 had originally suggested the following validation:

“Thus, at least some key TGE features (indicated in Figure 3) should be experimentally checked at independent, randomly-selected, IR sites in model cells and parental mES cells for establishing whether or not IR+ regions maintain the TGE features characteristic of the respective, pre-existent, native loci.”

Random selection would likely yield IRs in mostly silenced states, or with (subtle) variations in different combinations of epigenomic features. Assuming we would detect changes in an epigenomic feature between model (TRIP) and parental (wild-type) mES cells for some IRs, it would be difficult to determine the impact that the changes could have had on Cas9 mutation efficiency based on a few IRs. As an example, Daer et al. (2017) showed that Cas9 targets within 200bp with similar variations in a particular epigenomic feature could exhibit detectable or non-detectable effects on mutation efficiency, based on gRNA, and despite a clear effect on Cas9 binding. Please see a brief discussion of their findings at the end of this document.

We believe that this question needs extensive investigation beyond probing epigenomic features for a few IR loci. Please note that we do not dispute that TRIP integrations could induce changes to TGE features. However, the question in this context is whether those changes could be the main cause of the weak correlations observed between Cas9 mutation efficiency and epigenomic features. Given the challenges associated with such an assessment, and the agreement of our results with the literature on native epigenomic features and endogenous loci (Wu, 2014; Daer, 2017; Uusi-Mäkelä, 2018), we hope the Reviewers can agree to consider our analysis on epigenomic features as exploratory.

Our conclusion is that Cas9 mutation efficiency is clearly dependent on genomic location, while the effect of specific factors such as individual epigenomic features remains unclear and is not yet well understood.

Minor comments

It is somewhat odd the author's reply to the earlier minor comment 3 (and 4) to briefly describe key experimental steps.

Minor comment 3. To facilitate understanding of the methodology and interpretation of the results, briefly, described the methods indicated in: “Cell culture and TRIP library construction and transfection were performed according to the TRIP protocol12, 13.”

Author's answer: “We believe it is common practice to refer to the original work, rather than describing the protocol again. It is also not trivial to abbreviate the description of the protocol, as it might jeopardize reproducibility. Nevertheless, we are willing to include it if the Reviewer and Editor feel that this would be necessary.”

As the Reviewer quotes, the TRIP protocol is described in detail in (Akhtar et. Cell 2013) and (Akhtar et al., Nature Protocols 2014). We have been hesitant to republish previous methods, since it is common

practice to refer to previous work and we believe that republishing could be interpreted as self-plagiarism. We will follow the Editor's guidance whether we should or not incorporate this protocol again.

Reference: Wu, X. *et al.* Genome-wide binding of the CRISPR endonuclease Cas9 in mammalian cells. *Nat. Biotechnol.* **32**, 670 (2014).

Annex: Review of studies cited by Reviewer 1

Daer et al., ACS Synthetic Biology 2017

The authors use a transgenic cell line with a drug-inducible switch to control chromatin states (open and closed) at a single genomic locus, similar to the article we already cite (Chen et al., NAR 2016). We include the main result below (Fig. 3c), showing the mutation efficiency for each of 9 sgRNAs (denoted by sg0XY). Color indicates chromatin state: blue is unsilenced, green is partially silenced, and red is fully silenced. Out of 9 guide RNAs, 3 show an effect between unsilenced and fully/partially silenced chromatin (sg046, sg032, sg054), 2 show an effect only between unsilenced and fully silenced chromatin (sg034, sg044), and 4 do not show an effect (sg055, sg031, sg025, sg048). The authors also showed that changes between unsilenced and fully silenced states produced significant reductions in Cas9 binding for three sites/sgRNAs, which did not necessarily lead to reductions in Cas9 mutation efficiency (Fig. 3c and 5c).

Effects are modest and inconsistent across sgRNAs (Figure 3c below), even though they result from pronounced changes to chromatin accessibility induced by active manipulation (see Figures 3d, 5a, 5b in the original Daer et al. manuscript). Changes in chromatin significantly affected Cas9 binding (Figure 3c below), but did not necessarily lead to detectable changes in Cas9 mutation efficiency (Figure 5c below). The authors suggest that effects may further depend on sgRNA, resolution of epigenomic context, among others.

Figure 3c: Effect on Cas9 mutation efficiency (Daer et al., ACS Synthetic Biology 2017)

Figure 5c: Effect on dCas9 binding (Daer et al. ACS Synthetic Biology 2017)

Uusi-Mäkelä et al., PLoS ONE 2018

The authors use public data on gRNA mutation efficiency in zebrafish (CRISPRz database), as well as public data for: expression (RNA-seq), exon methylation (ChIP), H3K4me3 (ChIP-seq, and chromatin accessibility (ATAC-seq). We include the main results (Tables 1-2) below, reporting correlations between mutation efficiency and expression or epigenetic features for distinct developmental stages.

The correlations are low (Tables 1-2) - the authors themselves refer to them as weak. Only a few correlations for certain features and stages were considered significant (namely expression, chromatin accessibility). This is in line with our observations. Importantly, we believe that p-values were not corrected for multiple testing, since this is never mentioned. If the p-values were corrected, most would not survive the significance threshold.

Developmental stage	Spearman correlation	p-value
64-cell	0.190	0.006*
oblong-sphere	0.227	0.001*
50%-epiboly	0.187	0.007*
15-somite	0.210	0.002*
36hpf	0.230	0.001*
48hpf	0.182	0.008*
60hpf	0.188	0.006*
72hpf	0.131	0.058

<https://doi.org/10.1371/journal.pone.0196238.t001>

Table 1 (Uusi-Mäkelä et al., PLoS ONE 2018)

Chromatin feature	n	Developmental stage/Timepoint	Spearman correlation	p-value
Exon methylation	263	1-cell	0.115	0.063
		Mid blastula transition	0.107	0.084
H3K4me3	47	75-80% epiboly	0.263	0.074
Chromatin accessibility	263	4hpf	0.182	0.003*

<https://doi.org/10.1371/journal.pone.0196238.t002>

Table 2 (Uusi-Mäkelä et al., PLoS ONE 2018)

Jensen et al., FEBS Letters 2017

The authors investigate the effect of chromatin accessibility on mutation efficiency at 10 genomic loci, using 2 gRNAs per loci, in HEK293T cells using two Cas9 variants. We include the main result in Figures 1(B-D) below. In Fig. 1B, the loci are sorted according to DNase I sensitivity seen in Fig. 1C. In Fig. 1D, the loci are grouped into open or closed according to the same DNase I sensitivity. The authors report a trend in mutation efficiency that correlates with DNase I sensitivity. However, the 10 loci differ in target sequence. The authors do not correct for this confounding factor.

Figures 1B-1D (Jensen et al., FEBS Letters 2017)

In an attempt to address the confounder, the authors performed a plasmid-based assay to assess guide RNA efficiency without influence of chromatin state (Figures 2A and 2B below). The authors reported that there was no difference between the efficiencies of the guide RNAs targeting loci in the open vs. the closed chromatin group in the plasmid system. However, the plasmid assay (Fig.2) measured efficiency using GFP expression instead of percentage of indels by Sanger sequencing and TIDE used in the endogenous assay (Fig.1). Since the readouts are different, the data in Fig. 1-2 are not comparable, and the data from Fig.2 could not be used to normalize the data in Fig.1. In addition, GFP expression is not a good indicator of mutation efficiency compared to percentage of indels, as not all mutations result in expression changes. Depending on the most common type of mutation generated by the assay, GFP expression could grossly underestimate the actual mutation efficiency, which would largely favor the hypothesis being tested in this study.

Figures 2A-2B (Jensen et al., FEBS Letters 2017)

We identified several important issues:

- The number of loci targeted in the plasmid system was different: 4 extra loci were targeted.
- The GFP expression in the plasmid assay seems higher for loci found in open endogenous regions (Fig. 2B). This suggests that gRNAs targeting open and closed regions had different efficiencies to start with. Considering that GFP expression typically underestimates the effect, we believe that differences in mutation efficiency (percentage of indels) could be more pronounced than is apparent from Fig. 2B.
- The presence of two outliers: one in the open group when targeting endogenously (Chr9_O2, Fig. 1B-1C), pulling the distributions apart (Fig. 1C), another one in the closed group when targeting in the plasmid system (Chr22.1_C1, Fig. 2A), bringing the distributions closer together (Fig. 2B).

The limited number of loci combined with the above issues make the presented effect sizes and statistical significance results unreliable.